# GLNCD: Graph-Level Novel Category Discovery

**Bowen Deng**
Sun Yat-sen University
bowen.deng20@gmail.com

**Lele Fu**
Sun Yat-sen University

**Sheng Huang**
Sun Yat-sen University

**Tianchi Liao**
Sun Yat-sen University

**Jialong Chen**
Sun Yat-sen University

**Tao Zhang**
Sun Yat-sen University
zhangt358@mail.sysu.edu.cn

**Chuan Chen** *
Sun Yat-sen University
chenchuan@mail.sysu.edu.cn

## Abstract

Graph classification has long assumed a closed-world setting, limiting its applicability to real-world scenarios where new categories often emerge. To address this limitation, we introduce Graph-Level Novel Category Discovery (GLNCD), a new task aimed at identifying unseen graph categories without supervision from novel classes. We first adapt classical Novel Category Discovery (NCD) methods for images to the graph domain and evaluate these baseline methods on four diverse graph datasets curated for the GLNCD task. Our analysis reveals that these methods suffer a notable performance degradation compared to their image-based counterparts, due to two key challenges: (1) insufficient utilization of structural information in graph self-supervised learning (SSL), and (2) ineffective pseudo-labeling strategies based on ranking statistics (RS) that neglect graph structure. To alleviate these issues, we propose ProtoFGW-NCD, a framework consisting of two core components: ProtoFGW-CL, a novel graph SSL framework, and FGW-RS, a structure-aware pseudo-labeling method. Both components employ a differentiable Fused Gromov-Wasserstein (FGW) distance to effectively compare graphs by incorporating structural information. These components are built upon learnable prototype graphs, which enable efficient, parallel FGW-based graph comparisons and capture representative patterns within graph datasets. Experiments on four GLNCD benchmark datasets demonstrate the effectiveness of ProtoFGW-NCD.

## 1 Introduction

Graph neural networks [70, 10, 1, 6] have become important in modern machine learning, finding applications in diverse domains such as protein function prediction [55], malware detection [2], and social network analysis [71]. However, most existing methods assume either a closed-world setting [70, 68, 74], where all test categories are known during training, or an unsupervised setting, where no categories are known [43, 42, 12, 64, 22]. In real-world scenarios, this assumption often breaks down: new graph categories may emerge dynamically (e.g., novel protein structures or previously unseen software behaviors), requiring models to adapt to an open-world setting. Despite the importance of this challenge, the problem of discovering novel graph categories remains unexplored.

We introduce graph-level novel category discovery (GLNCD), a new task that extends graph classification to open-world scenarios. Different from node-level NCD [32, 34, 11] aiming to discover new

---

*Corresponding author.

39th Conference on Neural Information Processing Systems (NeurIPS 2025).

node categories, GLNCD is to discover novel graph classes (i.e., cluster unlabeled graphs) without explicit label supervision for these classes. In Section 3.1, we construct four diverse GLNCD datasets spanning various graph classification domains, including bioinformatics [55], program analysis [21], social networks [71], and graph-based image classification [15]. These datasets serve as benchmarks for evaluating GLNCD methods and highlight the unique challenges posed by graph data.

We adapt classical visual NCD methods (originally designed for image data) [78, 27, 80, 60] to the GLNCD setting. A typical visual NCD method is comprised of self-supervised learning (SSL), supervised learning on known classes, and category discovery. By replacing these components with the counterparts in graph domain (e.g, replacing SSL [23, 24, 5, 7, 9, 76] with Graph SSL [56, 75, 73, 66, 41, 72, 81, 82, 73, 12, 35]), we get the graph variants of visual NCD methods (detailed in Section 3.2). Surprisingly, our experiments (see Figure 1 and Table 2) reveal that these adapted methods perform significantly worse on graph data than on image data. Through systematic investigation presented in Section 4, we identify two key limitations contributing to this underperformance: 1) Existing graph-level SSL methods fail to fully exploit the structural information in graphs, resulting in suboptimal graph encoders. 2) The unlabeled sample pseudo-labeling strategy commonly used in visual NCD, ranking statistics (RS) [27], does not account for the structural information of graphs, leading to low-quality pseudo-labels for unlabeled graph samples.

To address the above limitations, we propose ProtoFGW-CL, a new graph SSL framework, and FGW-RS, a structure-aware pseudo-labeling method. They constitute our GLNCD method, ProtoFGW-NCD (Section 5), and both rely on the Fused Gromov-Wasserstein (FGW) distance [59, 39], which enables structural comparison between attributed graphs—going beyond readout vectors in baseline methods. However, computing pairwise FGW distances between two batches of graphs $\mathcal{G}_1$ and $\mathcal{G}_2$ with varying sizes is inefficient using the BAPG solver [39] from existing tools [20]. To address this, we design a differentiable BAPG layer that supports efficient parallel computation during both forward and backward passes (Section B). Our BAPG layer assumes that the second group of graphs has uniform size. To handle variable-sized input graphs, we introduce $K$ learnable prototype graphs $\mathcal{G}^{(B)}$ of identical size. We then compute the FGW distance matrix $\mathbf{V}_1 \in \mathbb{R}^{b_1 \times K}$ between $\mathcal{G}_1$ and $\mathcal{G}^{(B)}$, and $\mathbf{V}_2$ between $\mathcal{G}_2$ and $\mathcal{G}^{(B)}$. By comparing $\mathbf{V}_1$ and $\mathbf{V}_2$, we derive the pairwise relationships between $\mathcal{G}_1$ and $\mathcal{G}_2$, which are central to graph SSL and unlabeled graph pseudo-labeling. To make $\mathcal{G}^{(B)}$ more than just computational intermediaries, we treat them as learnable parameters following [5]. These prototypes are updated through BAPG layer during training, enabling them to capture representative structural patterns from the dataset, thereby benefiting graph SSL and pseudo-labeling.

**Contributions**: **1)** We introduce graph-level novel category discovery (GLNCD), a new task that extends graph-level classification to open-world scenarios, along with four diverse benchmark datasets spanning multiple domains. **2)** We adapt classical visual NCD methods to the graph domain and conduct a systematic analysis, revealing their limitations in sufficiently capturing structural information in graphs. **3)** To address these limitations, we propose ProtoFGW-NCD, a novel GLNCD method built upon a differentiable Fused Gromov-Wasserstein (FGW) distance. **4)** We design a parallel, differentiable BAPG layer for extremely efficient pairwise FGW distance computation.

## 2 Related Work

### 2.1 Visual Novel Category Discovery for Image Data

Novel Category Discovery (NCD) for image data has been well studied. AutoNovel [27] generates binary pairwise pseudo-labels for unknown samples using ranking statistics (RS), providing learning signals for the MLP-based clustering head. The quality of these pseudo-labels is closely tied to the representation quality output by the encoder. Incorporating contrastive loss [80] can enhance both representation and pseudo-label quality, thereby improving NCD performance. Exploiting multi-scale representation can improve pseudo-label reliability; for instance, DualRS [78] leverages both global and local branches to capture large-scale and fine-grained visual information, respectively. UNO [19] introduces a Sinkhorn-based approach to generate pseudo-labels without requiring pairwise comparisons. To better leverage knowledge from known categories, rKD [25] employs a fixed, supervised encoder with a known category head to constrain model outputs on known classes during the discovery training phase. Vaze et al. [61] and Cao et al. [4] propose a more challenging setting where the model must classify known-category samples and cluster unknown-category samples simultaneously during inference. While these methods perform well on regular image data, adapting

them to graph-structured data requires tailored modifications. We find that straightforward adaptations, such as replacing a CNN encoder with a GNN encoder, yield poor performance (see Figure 1 and Table 2), suggesting the need for more exploration.

## 2.2 Open World Graph Learning

Recent graph learning research has begun tackling open-world settings, where a node's class may be one that was unseen during training. In the node classification context, this gives rise to open-set recognition (OSR) problem: the model must classify nodes from known classes while identifying any nodes belonging to novel classes as "unknown." The seminal OpenWGL method [67] tackles this by learning uncertainty-aware node embeddings via a variational GNN, so that nodes from novel classes yield high predictive uncertainty and can be automatically rejected during inference. Hoffmann et al. [31] propose a meta-model that aggregates multiple confidence scores and employs a weakly-supervised thresholding strategy to decide when to label a node as unknown. Zhang et al. [77] propose to generate proxy nodes for unknown categories, tackling inductive node OSR problem. Beyond merely rejecting unknown nodes, NCD methods to cluster unlabeled nodes are proposed. ORAL [34] detects and remove the edges linking old and novel categories to alleviate the bias towards old categories, and uses multi–layer predictions to generate pseudo labels for unknown nodes. Hou et al. [32] consider a different NCD task that provides old-class nodes in the first training stage and then novel-class nodes in the second one. It is worth noting that open-world graph learning has been explored almost exclusively at the node level so far, and our work is the first to focus on graph-level.

# 3 Datasets and Baselines Adapted from Visual NCD

## 3.1 Prepare GLNCD datasets

Novel Category Discovery (NCD) can be viewed as a relaxed version of multi-class classification, where the goal is to group samples from the same novel category into the same cluster, without requiring them to be assigned to a specific label (e.g., $y = 0$). Therefore, one common approach to constructing an NCD dataset is to treat a subset of classes in a standard multi-class classification dataset as novel categories.

Although over 120 public graph-level datasets [50, 15] are available, we observe that only about five real-world multi-class graph classification datasets exist outside of computer vision scenarios. From these, we select three representative datasets spanning diverse domains, which, together with CIFAR10 (graph) [15], form the benchmark for GLNCD (Table 1). **Bioinformatics**: Graph structures naturally model protein molecules, where nodes represent secondary structure elements and edges indicate either sequential or spatial proximity between elements. The ENZYMES dataset [55, 50] contains six types of proteins with different catalytic functions. We treat the first three classes as old categories and the remaining three as novel categories. **Program Analysis**: The MalNet-Tiny dataset [21] consists of function call graphs (FCGs) derived from Android APK static disassembly. Each graph corresponds to a program type (Addisplay, Adware, Benign, Downloader, Trojan). We use the first three classes as old categories and the last two as novel categories. **Social Networks**: In the REDDIT12K dataset [71, 50], each graph represents a discussion thread, where nodes correspond to users and edges denote comment responses. The dataset contains 12 types of graphs, each corresponding to a subreddit. We designate the first six classes as old categories and the last five as novel categories. **Computer Vision**: An image can be decomposed into super-pixels, each forming a node whose features are computed as the average RGB values and spatial coordinates of the constituent pixels. Each node is connected to its eight nearest neighbors via edges weighted by Gaussian similarity. The CIFAR10 (graph) dataset [15] is constructed from CIFAR10 (image) [38] in this way. And we treat the first five classes as old categories and the rest as novel ones.

Luo et al. [46] incorporated various modern deep learning techniques, e.g., batch normalization [33] and residual connections [30], into GCN [37] and GIN [70] for graph-level tasks. More importantly, they employed Random Walk Structural Encoding (RWSE) [40, 16] as a preprocessing step to extract structural information, which is then used as part of the node and edge features. For all four datasets used in this work, we also apply RWSE-based preprocessing with a maximum path length of 32.

Table 1: Overview of GLNCD datasets. # steps is the maximal random walking length [46].

| Dataset | # graphs | Avg. # nodes | Avg. # edges | # node/edge feats | # steps | # old/new classes |
|---|---|---|---|---|---|---|
| ENZYMES | 600 | 32.6 | 124.3 | 21/0 | 32 | 3/3 |
| MalNet-Tiny | 5000 | 1410.3 | 2859.9 | 0/0 | 32 | 2/3 |
| REDDIT12K | 11929 | 391.41 | 456.89 | 0/0 | 32 | 5/6 |
| CIFAR10 | 60000 | 117.6 | 941.1 | 5/1 | 32 | 5/5 |

## 3.2 Design GLNCD Baselines with Visual NCD Methods

A number of NCD methods have been developed in the field of computer vision. If these can be directly adapted to the graph-level setting, it would offer a convenient solution to GLNCD. To explore this, we adapt three representative visual NCD methods, i.e., AutoNovel [27], NCL [80], and DualRS [78], to the graph domain. These models typically consist of an encoder $f_\theta$ and two classification heads: $h_n$ (for novel categories) and $h_o$ (for old known categories). The encoder is trained using SSL, the old-class head is trained with standard supervised signals from labeled samples, and the new-class head relies on specifically designed pseudo-labeling strategies for training. By replacing each component with its counterpart in the graph domain, we achieve a straightforward adaptation.

**AutoNovel**: (*Stage1*) Originally employs a RotNet approach [23] for pre-training, where the encoder learns to recognize rotation angles (0°, 90°, 180°, or 270°) to capture semantical image features. (*Stage 2*) After pretraining, the GNN encoder and the old-class head is trained on old-class samples. (*Stage 3*) Finally, the new-class head is trained using pairwise ranking statistics (RS) pseudo-labels generated with new-class sample representations. *Adaptation*: Since graph data consists of abstract topological structures without spatial orientation, we replace RotNet with GraphCL [75], a contrastive learning method tailored for graph-level tasks. Other stages remain unchanged.

**NCL**: Builds upon AutoNovel by introducing a MoCo-style [29] contrastive loss in (*Stage 3*), where negative samples are drawn from queues representing labeled and unlabeled data. Positive pairs include augmented views of the same sample and samples sharing the same old class label [36]. A hard-negative mining strategy is also incorporated [54]. *Adaptation*: The readout outputs of the encoder are Euclidean, so the changes of NCL from AutoNovel can directly fit into graph domain.

**DualRS**: (*Stage 1*) Abandons the RotNet-pretrained ResNet18 and instead uses a ResNet50 backbone pretrained on ImageNet [13] with MoCoV2 [8]. (*Stage 2*) is removed. (*Stage 3*) Both classification heads are equipped with global and local branches. The global branch captures coarse-grained representations, while the local branch focuses on fine-grained details. Each branch generates pseudo-labels via RS to train its respective head, and then distilled to the other branch. *Adaptation*: We use GraphCL to pretrain the GNN encoder. All other components remain unchanged.

We largely retain the original names of the visual NCD methods when referring to their adaptations for the graph domain. To avoid possible confusion in some cases, we prefix them, in these instances, with "G-" and refer to them as G-AutoNovel, G-NCL, and G-DualRS, respectively.

# 4 Challenges in NCD Method Adaptation: From Image to Graph Data

In this section, we reveal the failure of adapting the visual NCD methods to the graph domain like Section 3.2, and analyze the underlying reasons.

## 4.1 Why Direct Adaptation Fails? Ranking Statistics (RS) Fails

Table 2 shows the performance of AutoNovel on three image datasets, as well as its adapted version for graph-level tasks on four GLNCD datasets (Section 3.1), following the modifications described in Section 3.2. The New ACC (Train) on graph datasets, which is the primary metric of interest in NCD, is significantly lower than that on image datasets. More importantly, we observe a substantial performance gap between novel-class and old-class samples on graph datasets (Table 2), a phenomenon not present in image datasets. This suggests that the direct adaptation of AutoNovel fails to effectively leverage knowledge from old classes to aid in clustering unlabeled novel-class samples.

Given that pseudo-label quality plays a critical role in the performance of NCD methods, we hypothesize that the observed failure is due to the inability of RS to produce reliable pseudo-labels

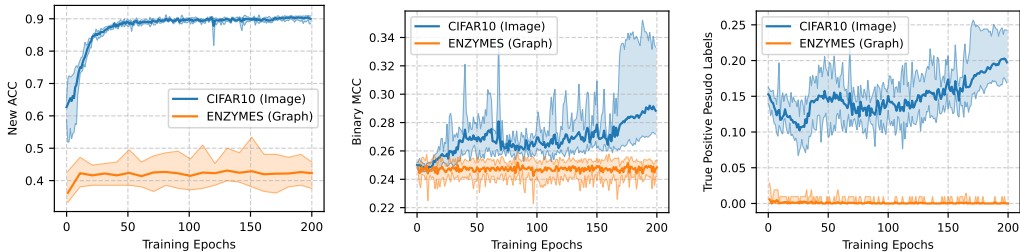

(a) The clustering accuracy on unlabeled new-class samples.
(b) The MCC (quality) of RS pseudo labels for sample pairs.
(c) The ratio of true positive RS pseudo labels for sample pairs.

Figure 1: Training dynamics of AutoNovel [27] on CIFAR10 (image) and ENZYMES (graph). **(a)** The performance on unlabeled training dataset. **(b)** The Matthews Correlation Coefficient (MCC) to evaluate the quality of pairwise pseudo-labels generated via ranking statistics (RS). **(c)** The ratio of samples for which at least one true positive (same-class) pair is identified by RS. The definitions of these two pseudo-label metrics and the rationale for their selection are provided in Appendix A.

Table 2: The average performance (over 10 runs) of AutoNovel [27] on image and graph datasets. Old ACC (Test) is the accuracy on the old-class samples in test dataset. New ACC (Train) is the clustering accuracy on the unlabeled training dataset. The last row is the gap between these metrics.

| Datasets | Image | | | Graph | | | |
|---|---|---|---|---|---|---|---|
| | CIFAR10 | CIFAR100 | SVHN | ENZYMES | MalNet-Tiny | REDDIT12K | CIFAR10 |
| Old ACC (Test) | 95.34 | 74.51 | 98.10 | 73.00 | 93.30 | 67.59 | 61.36 |
| New ACC (Train) | 88.50 | 74.28 | 94.21 | 41.90 | 74.51 | 39.21 | 41.67 |
| Old (row1) - New (row2) | 6.84 | 0.23 | 3.89 | 31.10 | 18.79 | 28.38 | 19.69 |

on graph-based datasets. To verify this, we track the New ACC (Train) along with RS pseudo-label quality on CIFAR-10 (image) and ENZYMES (graph), as shown in Figure 1. The results indicate that RS produces higher-quality pseudo-labels on CIFAR-10, with an overall increasing trend during training (Figure 1b and 1c). In contrast, on ENZYMES, the quality of pseudo-labels remain consistently low throughout training, suggesting that RS fails on graph-structured data in the direct adaptation described in Section 3.2.

## 4.2 Why RS Fails? Insufficient Exploration of Graph Structure

Ranking statistics (RS) generates pseudo-labels based on the representations output by the encoder (see Appendix A for the details of generation). If the encoder produces high-quality representations (i.e., samples from the same class are similar and those from different classes are dissimilar), the resulting RS pseudo-labels are typically of higher quality.

We therefore suspect that the representation learned by GraphCL in G-AutoNovel is not sufficiently discriminative. This hypothesis would be supported if higher-quality representations led to better pseudo-labels. However, according to recent reports [69, 26, 79], GraphCL performs as well or better than other graph SSL methods across a wide range of datasets, suggesting that replacing GraphCL with alternative SSL methods would not yield a significant improvement in representation quality for comparison purposes. To validate our hypothesis in another way, we pretrain three GNN encoders with increasing representation quality using the Supervised Contrastive (SupCon) loss [36], under settings where 0% (i.e., standard SSL), 50%, and 100% of the ground-truth binary pairwise labels are known. We then evaluate how the quality of RS pseudo-labels and NCD performance vary across these three levels of representation qualities.

Contrary to expectations, the quality of RS pseudo-labels (Figure 2b and 2c) shows little difference across the three representation qualities. Strikingly, despite nearly identical pseudo-label quality, the final NCD performance varies significantly across the three setups. **1)** This suggests that representation quality primarily influences the pseudo-label utilization rather than the pseudo-label quality itself: higher-quality representations provide a better data manifold, which facilitates learning better decision boundaries even when pseudo-labels are of comparable quality. **2)** This result does not imply that

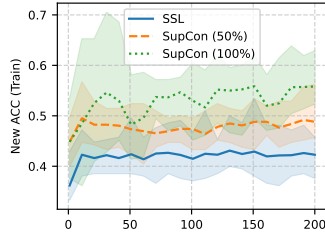 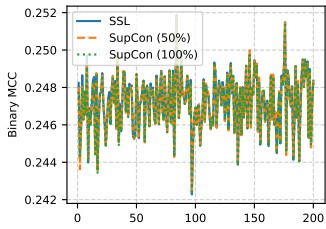 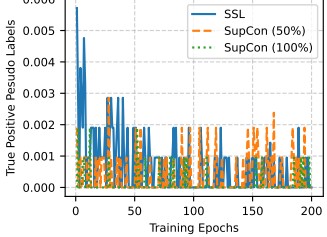

(a) The clustering accuracy on unla- (b) The MCC (quality) of RS pseudo (c) The raito of true positive RS
beled new-category samples.  labels for sample pairs.  pseudo labels for sample pairs.

Figure 2: Training dynamics of G-AutoNovel [27] on ENZYMES (graph). Three GIN encoders are pretrained with 0% (SSL), 50% (SupCon), and 100% (SupCon), true binary pairwise labels. **(a)** The performance on unlabeled training dataset. **(b)** The Matthews Correlation Coefficient (MCC) to evaluate the quality of RS pseudo-labels. **(c)** The ratio of samples for which at least one true positive pair is identified by RS. The details about of these pseudo-label metrics are provided in Appendix A.

representation quality has no impact on pseudo-label quality: as shown in Figure 1, on CIFAR-10 (image), RS pseudo-label quality improves substantially as the encoder learns better representations during training. Therefore, the nearly identical pseudo-label quality is likely due to the fact that RS is not well-suited to graph data and thus cannot effectively leverage improvements in representation quality to generate better pseudo-labels. We attribute the failure of RS to two principal factors: **1)** insufficient GNN pretraining that limits the representation quality and thus RS pseudo-label utilization; **2)** RS is unsuitable for graph data and fails to improve pseudo-label quality even when representation quality increases. Both issues stem from insufficient exploitation of graph structure:

- *Methods like GraphCL do not make sufficient use of structural information, resulting in suboptimal representations;*
- *RS operates solely on the readout graph representation vectors, thereby discarding part of valuable graph-structure information.*

## 5 Proposed Method: ProtoFGW-NCD

To test our hypothesis from Section 4.2—that the absence of structural information adversely affects both the SSL (i.e., GraphCL) and RS modules of AutoNovel—we developed ProtoFGW-NCD (Figure 3). This method was designed as a controlled experiment; it mirrors the architecture of AutoNovel exactly, with the sole exception of a Bregman Alternating Projected Gradient (BAPG) layer that injects structural information into the SSL and RS pipelines. Crucially, this integration requires no auxiliary loss functions or sophisticated mechanisms. This design provides a clean and direct means to validate our claim by allowing any performance difference to be attributed solely to the inclusion of structural information. The differentiable BAPG layer functions by solving the optimal transport problem between batches of attributed graphs [59, 65, 53], yielding rich structural information for graph comparison. With the inclusion of this layer, the original SSL and RS components of AutoNovel are adapted into our proposed ProtoFGW-CL and FGW-RS modules, respectively.

### 5.1 ProtoFGW-CL: Graph-level Representation Learning by Swapping Transport Couplings

GraphCL [75] utilizes graph structure only in the message-passing phase of the GNN encoder. The contrastive loss is computed solely on Euclidean vectors obtained via readout operations (e.g., min, sum), and does not explicitly compare structural differences between graphs $G_i$ and $G_j$ at this critical stage. Other graph-level SSL methods not only perform similarly to GraphCL but also fail to make full use of graph structure. For instance, some approaches generate views based on structural information (e.g., graph diffusion, communities, or motifs) [62, 57, 28, 58], while others employ multi-channel encoders to exploit more the original graph [69, 47]. However, like GraphCL, they all ignore structures when computing the distances between different graphs or graph views.

To address this, we propose ProtoFGW-CL, a graph-level contrastive learning method that explicitly computes the distances considering graph structures. Specifically, we compute the Fused Gromov-Wasserstein (FGW) distances between each graph and prototype graphs, normalizes it into codes, and enforces consistency across views. These randomly initialized prototype graphs $\left\{ G_k^{(B)} = (\mathbf{A}_k^{(B)}, \mathbf{Z}_k^{(B)}) | \mathbf{A}_k^{(B)} \in \mathbb{R}_{\geq 0}^{N_k \times N_k}, \mathbf{Z}_k^{(B)} \in \mathbb{R}^{N_k \times d_k} \right\}_{k=1}^{K}$ establish the coordinates for cross-view structure-aware graph comparisons, and are learned with backpropagation to capture representative graph patterns in dataset. The FGW distance (Definition 2) [59, 39] between two graphs is defined as the optimal transport between their probability measures given by Definition 1.

**Definition 1.** (Graph as Probability Measure) An attributed graph $G_1 = (\mathbf{C}_1, \mathbf{X}_1)$ with $N_1 = |V_1|$ nodes defines a metric-measure (MM) space $(V_1, \mathbf{C}_1, \mu_1)$. Each node $v_i$ has an explicit node feature vector $\mathbf{x}_{1,i} \in \Omega_x \subset \mathbb{R}^d$ and an implicit structural feature $\mathbf{s}_{1,i} \in \Omega_s$. The pairwise relationship among structural features is encoded in $\mathbf{C}_1$, where $C_{1,ij} = D_{\Omega_s}(\mathbf{s}_{1,i}, \mathbf{s}_{1,j})$ typically represented as an adjacency or Laplacian matrix. The probability measure associated with $G_1$ is defined as $\mu_1 = \sum_{i=1}^{N_1} h_i^{(1)} \delta_{(\mathbf{s}_{1,i}, \mathbf{x}_{1,i})}$, where $h^{(1)} \in \mathcal{H}_N = \left\{ h \mid h \in \mathbb{R}_{>0}^N, \sum_{i=1}^N h_i = 1 \right\}$ is a simplex histogram, and $\delta_{(\mathbf{s}_{1,i}, \mathbf{x}_{1,i})}$ denotes the Dirac delta function located at $(\mathbf{s}_{1,i}, \mathbf{x}_{1,i})$.

**Definition 2.** (FGW distance) Given two attributed graphs $G_1 = (\mathbf{C}_1, \mathbf{X}_1)$ and $G_2 = (\mathbf{C}_2, \mathbf{X}_2)$, their corresponding probability measures are $\mu_1 = \sum_{i=1}^{N_1} h_i^{(1)} \delta_{(\mathbf{s}_{1,i}, \mathbf{x}_{1,i})}$ and $\mu_2 = \sum_{j=1}^{N_2} h_j^{(2)} \delta_{(\mathbf{s}_{2,j}, \mathbf{x}_{2,j})}$, respectively. The FGW distance between them is defined as

$$\inf_{\pi \in \Pi} \left\{ \sum_{i,k=1}^{N_1} \sum_{j,l=1}^{N_2} \left[ (1-\alpha) D_{\Omega_x}(\mathbf{x}_{1,i}, \mathbf{x}_{2,j}) + \alpha |\mathbf{C}_1(i,k) - \mathbf{C}_2(j,l)|^2 \right] \mathbf{T}_{i,j} \mathbf{T}_{k,l} \right\}, \quad (1)$$

where $\Pi = \left\{ \mathbf{T} \in \mathbb{R}_{\geq 0}^{N_1 \times N_2} \mid \mathbf{T} \mathbf{1}_{N_2} = h^{(1)}, \mathbf{T}^\top \mathbf{1}_{N_1} = h^{(2)} \right\}$ is the feasible set of transport plans, $h^{(1)}$ and $h^{(2)}$ denote marginal distributions and are typically set to uniform distributions in practice, $D_{\Omega_x}$ is a metric in $\Omega_x$, and $\alpha \in [0, 1]$ balances the contributions of node features and graph structures.

## 5.2 FGW-RS: Ranking Statistics with More Graph Structure Information

G-AutoNovel applies ranking statistics (RS) to the Euclidean readout summary vectors, while graph structure is only implicitly used during message passing. Given the importance of structural information in graph-level tasks, this utilization is insufficient. To enable RS to more directly incorporate graph structure information, we construct two attributed graphs from the representations before readout $G_1^z = (\mathbf{A}_1, \mathbf{Z}_1)$ and $G_2^z = (\mathbf{A}_2, \mathbf{Z}_2)$, corresponding to the two graph samples. We then compute their FGW distances to a set of prototype graphs, resulting in distance vectors $\mathbf{v}_1, \mathbf{v}_2 \in \mathbb{R}_{\geq 0}^K$, which are subsequently fed into classic RS used in [27].

The prototype graphs, learned during training, capture diverse representative structural patterns in the dataset. The FGW distances $\mathbf{v}_1, \mathbf{v}_2$ explicitly encode both the input samples' structures and their relations to global patterns in the dataset. This allows RS to generate pseudo-labels that better reflect structural information, thereby improving the quality of pseudo-labels.

## 5.3 ProtoFGW-NCD: Integrating ProtoFGW-CL and FGW-RS

Unlike AutoNovel [27], which separates pretraining, supervised learning, and category discovery from each other, ProtoFGW-NCD performs representation learning, supervised learning and category discovery within a single training process. Figure 3 illustrates the architecture and pipelines. At the beginning of training, we randomly initialize a set of prototype graphs $\left\{ G_k^{(B)} = (\mathbf{A}_k^{(B)}, \mathbf{Z}_k^{(B)}) \right\}_{k=1}^K$. Labeled old-class samples and unlabeled new-class samples from the training set are mixed and randomly sampled into mini-batches. For each graph sample $G_i = (\mathbf{A}_i, \mathbf{X}_i, \mathbf{E}_i)$, $\mathbf{A}_i \in \mathbb{R}_{\geq 0}^{N_i \times N_i}$ is the adjacency matrix, $\mathbf{X}_i \in \mathbb{R}^{N_i \times d_n}$ is the node feature matrix, and $\mathbf{E}_i \in \mathbb{R}^{E_i \times d_e}$ is the edge feature matrix. **Data Augmentation**: We randomly remove $p\%$ of the nodes (and their edges) to get the subgraph $\tilde{G}_i = (\tilde{\mathbf{A}}_i, \tilde{\mathbf{X}}_i, \tilde{\mathbf{E}}_i)$ induced by the rest nodes. **Encoding**: Both $G_i$ and $\tilde{G}_i$ are passed through a learnable feature encoder to refine node and edge features, followed by a GNN+ encoder $f_\theta$ [46]. The outputs at the final GNN layer are denoted as $(\mathbf{A}_i, \mathbf{Z}_i)$ and $(\tilde{\mathbf{A}}_i, \tilde{\mathbf{Z}}_i)$, where $\mathbf{Z}_i$ and $\tilde{\mathbf{Z}}_i$ are the learned node representations. **FGW Codes**: We compute the FGW distances between these

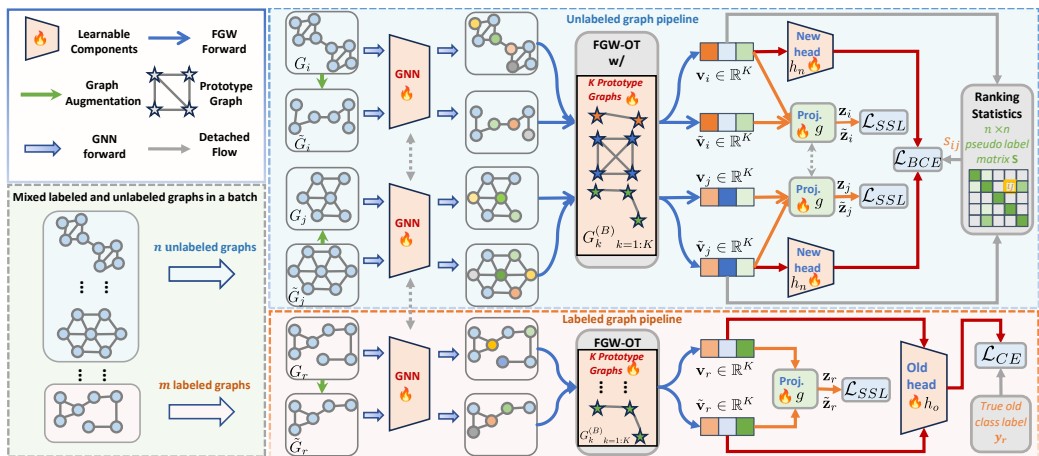

Figure 3: Illustration of **ProtoFGW-NCD**. Unlike previous graph SSL methods that directly compare Euclidean representations, **ProtoFGW-CL** maps graphs to codes by aligning them with learnable prototype graphs using the Fused Gromov-Wasserstein (FGW) distance. Under the supervision of $\mathcal{L}_{SSL}$, these codes are encouraged to be consistent across views, such that the code from one view can be predicted using the other. **FGW-RS**: The codes of $n$ unlabeled graphs are fed into RS to generate pseudo-label matrix $\mathbf{S}$, and also into the new-class head to obtain predictions. The pairwise similarities between these predictions are guided by $\mathbf{S}$ with $\mathcal{L}_{BCE}$. **Supervised Learning**: The codes of $m$ labeled graphs and their augmentations are fed into the old-class head to compute $\mathcal{L}_{CE}$.

outputs and the $K$ prototype graphs, resulting in distance vectors $\mathbf{v}_i, \tilde{\mathbf{v}}_i \in \mathbb{R}^K$. Let $m$ and $n$ denote the number of labeled and unlabeled samples in a batch, respectively. The overall training loss is composed of three terms

$$\mathcal{L} = \mathcal{L}_{CE} + \mathcal{L}_{BCE} + \mathcal{L}_{SSL} \tag{2}$$

$$\mathcal{L}_{CE} = -\frac{1}{2m} \sum_{r=1}^{m} \log \left[h_o(\mathbf{v}_r)\right]_{y_r} + \log \left[h_o(\tilde{\mathbf{v}}_r)\right]_{y_r} \tag{3}$$

$$\mathcal{L}_{BCE} = -\frac{1}{n^2} \sum_{i=1}^{n} \sum_{j=1}^{n} \left[ s_{ij} \log h_n \left(\mathbf{v}_i\right)^{\top} h_n \left(\mathbf{v}_j\right) + (1 - s_{ij}) \log \left(1 - h_n \left(\mathbf{v}_i\right)^{\top} h_n \left(\mathbf{v}_j\right)\right) \right] \tag{4}$$

$$\mathcal{L}_{SSL} = \frac{1}{2(m+n)} \sum_{i=1}^{m+n} \sum_{k=1}^{K} [\mathrm{SM}(\tilde{\mathbf{z}}_i)]_k \log \left([\mathrm{SM}(\mathbf{z}_i/\tau)]_k\right) + [\mathrm{SM}(\mathbf{z}_i)]_k \log \left([\mathrm{SM}(\tilde{\mathbf{z}}_i/\tau)]_k\right) \tag{5}$$

where $\mathrm{SM}(\cdot)$ is the softmax operator, $g(\cdot)$ is the projector for SSL, $\tilde{\mathbf{z}}_i = g(\tilde{\mathbf{v}}_i)$, $\mathbf{z}_i = g(\mathbf{v}_i)$, $\mathcal{L}_{CE}$ provides supervised signals from old-class samples, $\mathcal{L}_{BCE}$ guides the new-class head $h_n$, and $\mathcal{L}_{SSL}$ encourages the cross-view consistency of FGW codes. Here, $s_{ij} \in \{0, 1\}$ is the binary pseudo-label generated by ranking statistics (RS) based on structure-aware codes $\mathbf{v}_i$ and $\mathbf{v}_j$, and $s_{ij} = 1$ if RS considers $i$ and $j$ to belong to the same class. FGW is typically solved using iterative algorithms. If backpropagation is naively handled by PyTorch, it would record many intermediate steps, leading to an excessively large and computationally intractable computational graph. To address this, we design a parallel, differentiable FGW module, BAPG layer. This layer supports parallel differentiable FGW distances between $b_1$ `torch_sparse.SparseTensor` [18] and $b_2$ dense `torch.Tensor` on GPU. In contrast, previous differentiable FGW module [63] only supports CPU backend and relies on slow nested loops for pairwise FGW solving.

## 6 Experiments

### 6.1 Experimental Setup

We adopt the latest improved versions of two classic GNN architectures, i.e., GCN⁺ and GIN⁺ [46], as the backbones for all GLNCD methods. These models achieve or approach state-of-the-art

Table 3: GLNCD results with GIN⁺ encoder. The 1st and 2nd results are highlighted.

| | ENZYMES | | MalNet-Tiny | | REDDIT12K | | CIFAR10 | | Avg. Rank ↓ | | |
| | Old ACC | New ACC | Old ACC | New ACC | Old ACC | New ACC | Old ACC | New ACC | Old ACC | New ACC | All |
|---|---|---|---|---|---|---|---|---|---|---|---|
| K-means | 50.33±5.19 | 39.90±4.01 | 54.75±0.00 | 40.24±0.11 | 35.22±1.58 | 30.56±2.04 | 42.16±3.52 | 40.76±4.18 | 5.00 | 4.00 | 4.500 |
| AutoNovel | 73.00±1.39 | 41.90±1.62 | 90.75±2.66 | 68.93±6.29 | 67.59±0.77 | 39.21±0.92 | 61.36±3.95 | 41.67±1.90 | 3.00 | 2.75 | 2.875 |
| NCL | 70.00±2.04 | 45.81±5.82 | 92.55±1.14 | 69.76±7.44 | 70.11±0.75 | 36.57±1.59 | 67.54±1.24 | 38.21±2.01 | 2.25 | 3.00 | 2.625 |
| DualRS | 76.33±0.75 | 39.52±1.12 | 92.85±1.05 | 69.09±3.37 | 66.05±0.58 | 39.68±1.77 | 65.78±1.39 | 40.23±0.66 | 2.25 | 3.25 | 2.750 |
| Ours | 74.00±5.68 | 37.95±3.16 | 93.30±0.76 | 74.51±7.05 | 67.39±0.48 | 39.94±2.42 | 56.44±1.43 | 44.04±1.58 | 2.50 | 2.00 | **2.250** |

Table 4: GLNCD results with GCN⁺ encoder. The 1st and 2nd results are highlighted.

| | ENZYMES | | MalNet-Tiny | | REDDIT12K | | CIFAR10 | | Avg. Rank ↓ | | |
| | Old ACC | New ACC | Old ACC | New ACC | Old ACC | New ACC | Old ACC | New ACC | Old ACC | New ACC | All |
|---|---|---|---|---|---|---|---|---|---|---|---|
| K-means | 39.67±1.83 | 38.86±2.89 | 66.60±9.26 | 58.72±2.76 | 42.34±4.28 | 37.05±2.11 | 42.27±0.12 | 40.72±1.42 | 5.00 | 3.75 | 4.375 |
| AutoNovel | 71.33±2.74 | 41.52±1.86 | 80.30±6.79 | 62.43±5.40 | 68.91±0.33 | 39.08±1.34 | 61.10±3.12 | 41.26±1.31 | 3.00 | 2.00 | 2.500 |
| NCL | 67.67±1.49 | 39.71±3.77 | 85.50±2.35 | 62.23±1.68 | 69.35±1.11 | 37.01±1.20 | 70.63±0.46 | 39.64±0.82 | 2.25 | 3.50 | 2.875 |
| DualRS | 64.67±3.21 | 39.33±5.07 | 68.75±7.45 | 49.53±3.74 | 66.47±0.87 | 40.76±2.77 | 70.90±0.86 | 39.17±0.86 | 3.50 | 3.75 | 3.625 |
| Ours | 72.17±5.67 | 44.84±3.07 | 80.95±6.16 | 63.35±1.19 | 69.43±3.74 | 40.81±2.16 | 71.25±0.61 | 38.92±0.49 | 1.25 | 2.00 | **1.625** |

performance on graph-level supervised tasks [46]. In the Fused Gromov-Wasserstein (FGW) distance Eq. (1), we use the Euclidean distance as node feature comparison metric $D_{\Omega_x}$, and the adjacency matrix $\mathbf{A}$ to represent pairwise structural relationship $\mathbf{C}$. The attributed graphs used in computing FGW distances are constructed by the node hidden features $\mathbf{Z}$ of the final GNN layer and the input graph adjacency matrix $\mathbf{A}$. Inspired by [63], we set the trade-off parameter $\alpha$ as learnable. For data graphs, the marginal probability distributions are set to uniform, while for prototype graphs the marginals are made learnable. ProtoFGW-CL aims to incorporate structural information into the contrastive loss. So for graph data augmentation, we follow GraphCL and simply apply random node dropping: a fraction $p\%$ of nodes are randomly removed, along with all edges connected to them.

## 6.2 Main Results

The GLNCD datasets constructed in Section 3.1 each include a training set, validation set, and test set, as detailed in Table 7. We train the model on all labeled old-class samples and unlabeled new-class samples from the training set. During evaluation, we report two performance metrics: the clustering accuracy on unlabeled (new-class) training samples, denoted as **New ACC** and the classification accuracy on old-class test samples, denoted as **Old ACC**. We report the GLNCD results using a GIN⁺ encoder in Tables 3 while the results for the GCN⁺ encoder are provided in Table 4. As shown in these tables, ProtoFGW-NCD, which incorporates more graph structural information, demonstrates decent NCD performance and achieves the highest average ranking on all datasets. This confirms the effectiveness of exploiting more graph structure in GLNCD tasks.

## 6.3 Ablation Study

Table 5: Ablation with GCN⁺ on ENZYMES

| Method | Old ACC | New ACC |
|---|---|---|
| wo FGW-RS | 72.00±5.14 | 33.33±0.00 |
| wo ProtoFGW-CL | 70.50±3.69 | 42.10±4.52 |
| wo Supervised learning | 34.00±2.74 | 41.05±3.36 |
| fixed proto. graphs | 62.33±4.32 | 38.19±2.12 |
| ProtoFGW-NCD | 72.17±5.67 | 44.84±3.07 |

We conduct ablation studies on the components of ProtoFGW-NCD by reporting both Old ACC and New ACC to evaluate their effectiveness. ProtoFGW-NCD consists of four key components: ProtoFGW-CL, FGW-RS, supervised learning, and learnable prototype graphs. Removing the first three components can be achieved by excluding $\mathcal{L}_{SSL}$, $\mathcal{L}_{BCE}$, and $\mathcal{L}_{CE}$ from the training loss. To assess the importance of adaptively learned prototype graphs, we disable gradient updates for them.

The results in Table 5 demonstrate that each component contributes significantly, and removing any leads to noticeable degradation. Among the components, FGW-RS has a minor impact on Old ACC but plays a decisive role in New ACC, as its removal leads to a drop to 33.33%, equivalent to random guessing among the three unlabeled categories. In contrast, ProtoFGW-CL significantly affects Old ACC while having minimal influence on New ACC. This may be because training the encoder and new head with pseudo-labels from FGW-RS itself serves as a form of self-supervised learning (SSL) beneficial for novel category discovery. Without supervised signals, although Old ACC degrades to the level of random guessing, New ACC still achieves notable performance, indicating that ProtoFGW-CL effectively guides the model to learn meaningful SSL representations for NCD.

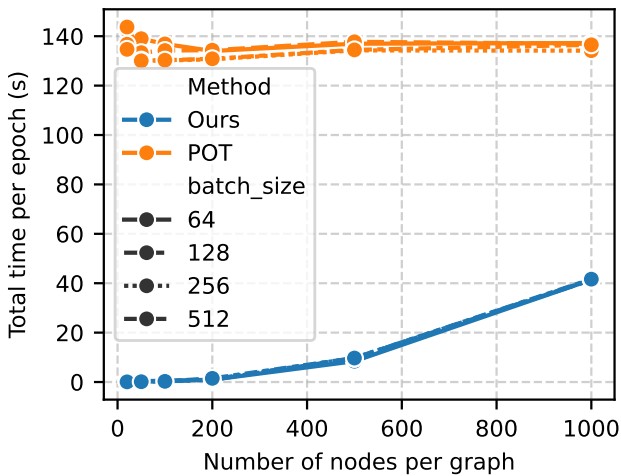

Figure 4: The epoch time (s) of different BAPG solver implementations for FGW problems. The x-axis indicates different synthetic datasets, i.e., CSBM-20-10, CSBM-50-10, ..., CSBM-1000-10.

Table 6: The average batch time (s) of different BAPG implementations. ↑ indicates speedup factor. The highest speedup for each dataset (row) across all batch sizes is shown in boldface.

| Batch Size $B$ | 64 | | | 128 | | | 256 | | | 512 | | |
|---|---|---|---|---|---|---|---|---|---|---|---|---|
| Dataset | POT | Ours | ↑ | POT | Ours | ↑ | POT | Ours | ↑ | POT | Ours | ↑ |
| CSBM-20-10 | 8.90 | 0.04 | 250.7 | 17.96 | 0.02 | 719.6 | 34.19 | 0.03 | 1296.8 | 67.34 | 0.03 | **2070.2** |
| CSBM-50-10 | 8.69 | 0.03 | 333.5 | 16.67 | 0.03 | 598.8 | 32.57 | 0.04 | 844.8 | 65.02 | 0.06 | **1020.1** |
| CSBM-100-10 | 8.55 | 0.03 | 289.3 | 16.78 | 0.04 | 416.9 | 32.54 | 0.07 | 457.6 | 65.14 | 0.14 | **474.6** |
| CSBM-200-10 | 8.38 | 0.06 | 142.6 | 16.79 | 0.11 | **150.2** | 32.66 | 0.28 | 117.9 | 65.43 | 0.74 | 88.0 |
| CSBM-500-10 | 8.56 | 0.53 | 16.2 | 17.22 | 1.20 | 14.3 | 33.56 | 2.43 | 13.8 | 67.23 | 4.85 | 13.8 |
| CSBM-1000-10 | 8.58 | 2.59 | **3.3** | 17.12 | 5.17 | 3.3 | 33.54 | 10.35 | 3.2 | 68.26 | 20.82 | 3.3 |

Learnable prototypes are critical for both Old ACC and New ACC, resulting in performance changes of approximately 10 and 6 points, respectively. This highlights the necessity of learnable prototypes and our efficient BAPG layer.

## 6.4 The Efficiency of Our BAPG Layer

In contrast to the BAPG operator in POT [20] which requires a brute-force loop, our proposed BAPG layer (Appendix B) can compute the Fused Gromov-Wasserstein (FGW) distance between $B_1$ and $B_2$ graphs in parallel. We compare their efficiency on Contextual Stochastic Block Model (CSBM) graphs [14, 48] with varying numbers of nodes. For a graph dataset with N nodes, we assume there are 10 prototype graphs, each with $\frac{N \log N}{2}$ nodes (see Appendix C.2 for more details) and denote by CSBM-N-10 the synthesized dataset. The computation times for FGW distance under different batch sizes are presented in Table 6 and Figure 4, where our method achieves a speedup of up to 2070.2x compared to the POT implementation.

## 7 Conclusion

This paper introduces Graph-Level Novel Category Discovery (GLNCD) task, aiming to identify unseen graph categories in an open-world setting. We present four diverse benchmark datasets across different scenarios and systematically adapt classical NCD methods for images to the graph domain. However, experimental results show that these direct adaptations perform poorly on graph data due to insufficient graph structure explorations. To address this issue, we propose ProtoFGW-NCD that consists of structure-aware SSL and pseudo-labeling, via a differentiable Fused Gromov-Wasserstein (FGW) module, BAPG layer. Experiments demonstrate that ProtoFGW-NCD matches or outperforms baseline methods. As a direction for future research, it would be promising to develop methods that explore graph structures more efficiently than FGW module for GLNCD tasks on large-scale datasets.

## Acknowledgments

The research is supported by the National Key R&D Program of China (2023YFB2703700), the National Natural Science Foundation of China (62176269).

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

# A  Measure the Quality of Pairwise Pseudo Labels for Unlabeled Samples

**The pseudo labels from ranking statistics (RS)**    Methods such as AutoNovel [27] generate pairwise pseudo-labels using ranking statistics (RS). Instead of directly computing the similarity (e.g., cosine similarity) between $d$-dimensional readout vectors $\mathbf{v}_i = \mathrm{Readout}(f_\theta(G_i))$ and $\mathbf{v}_j = \mathrm{Readout}(f_\theta(G_j))$ of samples $i$ and $j$, RS ranks the $d$ elements of each vector based on the element magnitudes. If the element ranking orders of two readout vectors are consistent, the corresponding samples are likely to belong to the same novel category, and we assign a pseudo-label $s_{ij} = 1$. Otherwise, we set $s_{ij} = 0$. The new-class head $h_n$ is then trained using binary cross-entropy loss and the pairwise binary labels:

$$\mathcal{L}_{\mathrm{BCE}} = -\frac{1}{M^2} \sum_{i=1}^{M} \sum_{j=1}^{M} \left[ s_{ij} \log h_n\left(\mathbf{v}_i\right)^\top h_n\left(\mathbf{v}_j\right) + (1 - s_{ij}) \log\left(1 - h_n\left(\mathbf{v}_i\right)^\top h_n\left(\mathbf{v}_j\right)\right) \right], \quad (6)$$

where $h_n\left(\mathbf{v}_i\right)$ is the softmaxed prediction on sample $i$ and $M$ is the number of unlabeled samples involved.

Han et al. [27] found the loss Eq. 6 to be the most critical component for NCD. Therefore, we aim to quantify its quality to enable deeper analysis. Given $M$ unlabeled new-class samples, RS generates a pseudo-label matrix $\mathbf{S} \in \{0,1\}^{M \times M}$. Let $\mathbf{T}$ denote the ground-truth pairwise label matrix, where $T_{ij} = 1$ if samples $i$ and $j$ belong to the same novel class, and $T_{ij} = 0$ otherwise. Evaluating the quality of RS can thus be framed as a binary classification problem.

**Evaluate pseudo labels with Matthews Correlation Coefficient**    According to Proposition 3, there are much more zeros than ones in $\mathbf{T}$. So the RS pseudo label quality evaluation is an imbalanced binary classification task, making accuracy an unsuitable evaluation metric. Instead, we adopt the Matthews Correlation Coefficient (MCC) [49], a widely used measure for evaluating classification performance under class imbalance. Let the confusion matrix for the binary classification be

$$\begin{pmatrix} TP & FN \\ FP & TN \end{pmatrix}.$$

Then MCC is defined as

$$MCC = \frac{TP \cdot TN - FP \cdot FN}{\sqrt{(TP + FP)(TP + FN)(TN + FP)(TN + FN)}},$$

which takes into account all four components of the confusion matrix, i.e., true positives (TP), true negatives (TN), false positives (FP), and false negatives (FN), and therefore provides a comprehensive measure of the agreement between generated pseudo-labels and the ground truth pairwise labels.

**Proposition 3.** *Consider a class-balanced dataset with $C$ categories, each containing $M$ samples. Define the pairwise comparison matrix $\mathbf{T} \in \{0,1\}^{CM \times CM}$ such that $T_{ij} = 1$ if samples $i$ and $j$ belong to the same class, and $T_{ij} = 0$ otherwise. Then, the fraction of positive entries (i.e., entries equal to 1) in $\mathbf{T}$ is given by*

$$\frac{CM^2}{(CM)^2} = \frac{1}{C}.$$

**Evaluate pseudo labels with the portion of samples with at least one same-class pair**    As stated in Proposition 3, the majority of sample pairs are negative, i.e., with a label 0. Each sample encounters far fewer true positive pairs than negative ones, making true positive pseudo-labels more valuable. Therefore, we use the ratio of samples that have encountered at least one true positive pair to quantify the quality of the pairwise pseudo-labels (generated by RS) .

# B  Parallel Differentiable BAPG Layer for Efficient FGW Distance

## B.1  Forward

For two attributed graphs $G_1 = (\mathbf{C}_1, \mathbf{X}_1)$ and $G_2 = (\mathbf{C}_2, \mathbf{X}_2)$, the Fused Gromov Wasserstein distance (Definition 2) between them amounts to, given $\mathbf{M} \in \mathbb{R}^{n_s \times n_t}$ defined by $M_{ij} = D_{\Omega_x}(\mathbf{x}_{1,i}, \mathbf{x}_{2,j})$,

the following quadratic optimization problem

$$FGW = \min_{\mathbf{T}} \quad (1-\alpha)\langle \mathbf{T}, \mathbf{M}\rangle_F + \alpha \sum_{i,j,k,l} L(\mathbf{C}_{1i,k}, \mathbf{C}_{2j,l})\mathbf{T}_{i,j}\mathbf{T}_{k,l}$$

$$s.t. \quad \mathbf{T} \in [0,1]^{n_s \times n_t} \qquad \mathbf{T1} = \mathbf{p} \qquad \mathbf{T}^T \mathbf{1} = \mathbf{q}.$$

According to Proposition 1 in [53], it holds that

$$\sum_{i,j,k,l} L(\mathbf{C}_{1i,k}, \mathbf{C}_{2j,l})\mathbf{T}_{i,j}\mathbf{T}_{k,l} = \left\langle \left[ \mathbf{C}_1^{\odot 2}\mathbf{p}\mathbf{1}_{N_2}^\top + \mathbf{1}_{N_1}\mathbf{q}^\top \left(\mathbf{C}_2^{\odot 2}\right)^\top - 2\mathbf{C}_1\mathbf{T}\mathbf{C}_2^\top \right], \mathbf{T}\right\rangle_F,$$

where $\mathbf{C}_1^{\odot 2}$ is the elementwise square and $\mathbf{1}_{N_2} \in \mathbb{R}^{n_t}$ is the all-one vector. Then we have

$$FGW = \min_{\mathbf{T}} \quad (1-\alpha)\langle \mathbf{T}, \mathbf{M}\rangle_F + \alpha \left\langle \left[ \mathbf{C}_1^{\odot 2}\mathbf{p}\mathbf{1}_{N_2}^\top + \mathbf{1}_{N_1}\mathbf{q}^\top \left(\mathbf{C}_2^{\odot 2}\right)^\top - 2\mathbf{C}_1\mathbf{T}\mathbf{C}_2^\top \right], \mathbf{T}\right\rangle_F$$

$$s.t. \quad \mathbf{T} \in [0,1]^{n_s \times n_t} \qquad \mathbf{T1} = \mathbf{p} \qquad \mathbf{T}^T \mathbf{1} = \mathbf{q}.$$

We take $L(\mathbf{C}_{1i,k}, \mathbf{C}_{2j,l}) = (\mathbf{C}_{1i,k} - \mathbf{C}_{2j,l})^2$ and employ the Bregman Alternating Projected Gradient (BAPG) method proposed in [39] to solve the optimal transport plan $\mathbf{T}$ and the corresponding FGW distance. The pseudo code of BAPG can be found in Algorithm 1.

---

**Algorithm 1:** POT BAPG Solver [39] for Fused Gromov-Wasserstein Distance (Forward)

---

**Input:** Node feature cost matrix $\mathbf{M} \in \mathbb{R}^{n_s \times n_t}$. Sparse structure matrices $\mathbf{C}_1 \in \mathbb{R}^{n_s \times n_s}$.
$\mathbf{C}_2 \in \mathbb{R}^{n_t \times n_t}$. Distributions $\mathbf{p} \in \mathbb{R}^{n_s}$, $\mathbf{q} \in \mathbb{R}^{n_t}$. Trade-off $\alpha \in (0,1)$, entropy $\epsilon > 0$.
Max iterations $T$, Tolerance $tol$
**Output:** Optimal transport matrix $\mathbf{T} \in \mathbb{R}^{n_s \times n_t}$ and $FGW$ distance

**1** Initialize $\mathbf{T} \leftarrow \mathbf{pq}^\top$;
**2** Elementwise function: $f(\mathbf{C}_1) \leftarrow \mathbf{C}_1^{\odot 2}, h(\mathbf{C}_1) \leftarrow \mathbf{C}_1, f(\mathbf{C}_2) \leftarrow \mathbf{C}_2^{\odot 2}, h(\mathbf{C}_2) \leftarrow \mathbf{C}_2$;
**3** Precompute constants: $R \leftarrow f(\mathbf{C}_1)\mathbf{p}\mathbf{1}_{n_t}^\top + \mathbf{1}_{n_s}(f(\mathbf{C}_2)\mathbf{q})^\top$;
**4** Define gradient operator $\nabla_{\mathbf{T}} \leftarrow 2\alpha(R - 2h(\mathbf{C}_1)\mathbf{T}h(\mathbf{C}_2)^\top + (1-\alpha)\mathbf{M}$;
**5 for** $t \leftarrow 1$ **to** $T$ **do**
**6** $\quad$ $\mathbf{T}_{\text{prev}} \leftarrow \mathbf{T}$;
**7** $\quad$ **Row projection:**
**8** $\quad\quad$ Update: $\mathbf{T} \leftarrow \mathbf{T} \odot \exp\left(-\nabla_{\mathbf{T}}/\epsilon\right)$
**9** $\quad\quad$ Normalize rows: $\mathbf{T}_{a,:} \leftarrow \frac{p_a}{\sum_j \mathbf{T}_{a,b}}\mathbf{T}_{a,:}$
**10** $\quad$ **Column projection:**
**11** $\quad\quad$ Update: $\mathbf{T} \leftarrow \mathbf{T} \odot \exp\left(-\nabla_{\mathbf{T}}/\epsilon\right)$
**12** $\quad\quad$ Normalize columns: $\mathbf{T}_{:,b} \leftarrow \frac{q_b}{\sum_a \mathbf{T}_{a,b}}\mathbf{T}_{:,b}$
**13** $\quad$ **if** $t \mod 10 = 0$ **then**
**14** $\quad\quad$ Compute error: $err \leftarrow \|\mathbf{T} - \mathbf{T}_{\text{prev}}\|_F$;
**15** $\quad\quad$ **if** $err < tol$ **then**
**16** $\quad\quad\quad$ **break**;
**17** $\quad\quad$ **end**
**18** $\quad$ **end**
**19 end**
**20 return** $\mathbf{T}$ and $FGW$;

---

Our proposed method, ProtoFGW-NCD, requires computing pairwise FGW distances between $b_1$ sparse adjacency matrices of varying sizes (represented as `torch_sparse.SparseTensor`) and $b_2$ dense adjacency matrices of prototype graphs with uniform size. Ideally, we aim to solve these FGW distances in parallel. However, both the official implementation [2] and the POT implementation [3] only support solving the FGW distance between one pair of dense matrices at one time. When $b_1$ and $b_2$ are large, invoking these implementations $b_1b_2$ times within nested loops results in prohibitively long computational time.

---

[2] https://github.com/squareRoot3/Gromov-Wasserstein-for-Graph
[3] https://pythonot.github.io/gen_modules/ot.gromov.html#ot.gromov.BAPG_fused_gromov_wasserstein

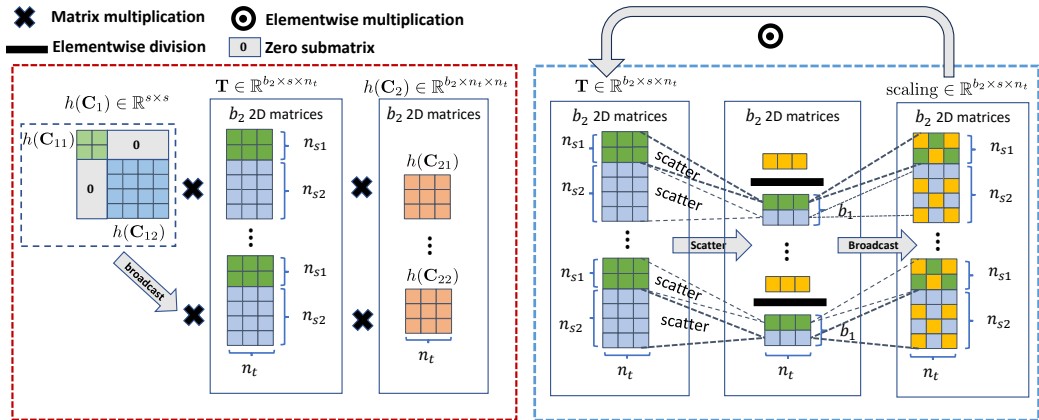

Figure 5: The illustration of two key operators in the forward of our BAPG layer, which parallels POT BAPG solver. The left is to parallel $h(\mathbf{C}_{1i})\mathbf{T}_{ij}h(\mathbf{C}_{2j})^\top$ (**Line 4** in Algorithm 1) and the right is to parallel $[\mathbf{T}_{ij}]_{:,b} \leftarrow \frac{q_b}{\sum_a [\mathbf{T}_{ij}]_{a,b}} [\mathbf{T}_{ij}]_{:,b}$(**Line 12** in Algorithm 1), for $i \in [1:b_1]$ and $j \in [1:b_2]$.

Observing Algorithm 1, we find that the main challenge in parallelizing the fused Gromov-Wasserstein (FGW) computation across $b_1 b_2$ groups lies in the irregular sizes of the $b_1$ sparse matrices. To address this, we propose constructing a sparse giant graph $\mathbf{C}_1 \in \mathbb{R}^{s \times s}$ (where $s = \sum_{i=1}^{b_1} n_{si}$) by arranging the sparse matrices $\mathbf{C}_{11}, \mathbf{C}_{12}, \ldots, \mathbf{C}_{1b_1}$ as diagonal blocks:

$$
\mathbf{C}_1 = \begin{bmatrix} \mathbf{C}_{11} & \mathbf{0} & \cdots & \mathbf{0} \\ \mathbf{0} & \mathbf{C}_{12} & \cdots & \vdots \\ \vdots & \vdots & \ddots & \vdots \\ \mathbf{0} & \mathbf{0} & \cdots & \mathbf{C}_{1b_1} \end{bmatrix} \in \mathbb{R}^{s \times s}.
$$

This allows for intra-batch parallelization even under irregular matrix dimensions. Meanwhile, the $b_2$ dense matrices $\mathbf{C}_{21}, \mathbf{C}_{22}, \ldots, \mathbf{C}_{2b_2}$ are stacked into a tensor $\mathbf{C}_2 \in \mathbb{R}^{b_2 \times n_t \times n_t}$, enabling parallelization along the first dimension via broadcasting rules. This design enables the $b_1 b_2$ pairs of FGW computations to be efficiently implemented using existing operators from `torch_sparse` and `torch_scatter`.

Since we need to solve for $b_1 b_2$ transport plans $\{\mathbf{T}_{ij} \in \mathbb{R}^{n_{si} \times n_t}\}, i \in [1:b_1], j \in [1:b_2]$, we represent them using a whole tensor $\mathbf{T} \in \mathbb{R}^{b_2 \times s \times n_t}$. Placing the batch size $b_2$ of dense matrices along the first dimension facilitates efficient sparse-dense matrix multiplication (via `spmm`) over all $b_2$ matrices. Our goal is to parallelize $b_1 b_2$ FGW with the operations among $\mathbf{C}_1 \in \mathbb{R}^{s \times s}$, $\mathbf{C}_2 \in \mathbb{R}^{b_2 \times n_t \times n_t}$, and $\mathbf{T} \in \mathbb{R}^{b_2 \times s \times n_t}$. The key computational steps in Algorithm 1 to parallelize are

- **Line 4** $h(\mathbf{C}_{1i})\mathbf{T}_{ij}h(\mathbf{C}_{2j})^\top$ that can be parallelized using the `spmm` operator from `torch_sparse`[4]

- **Line 12** $[\mathbf{T}_{ij}]_{:,b} \leftarrow \frac{q_b}{\sum_a [\mathbf{T}_{ij}]_{a,b}}[\mathbf{T}_{ij}]_{:,b}$ that requires computing the normalization sums $\sum_a [\mathbf{T}_{ij}]_{a,b}$ for each pair $(\mathbf{C}_{1i}, \mathbf{C}_{2j})$ using `scatter_sum` from `torch_scatter`[5], followed by grouped broadcasting across the first two dimensions of $\mathbf{T} \in \mathbb{R}^{b_2 \times s \times n_t}$.

We illustrate the above two parallelization designs in Figure 5. The forward process of our parallel BAPG layer is concluded in Algorithm 2, in a pytorch style.

---

[4]https://github.com/rusty1s/pytorch_sparse

[5]https://github.com/rusty1s/pytorch_scatter

**Algorithm 2: Our BAPG layer** for Fused Gromov-Wasserstein Distance (Forward)

**Input:** (1) Node feature cost matrix $\mathbf{M} \in \mathbb{R}^{b_2 \times b_1 \times n_s \times n_t}$. (2) Diagonal block matrix $\mathbf{C}_1 \in \mathbb{R}^{s \times s}$ stacked by $b_1$ sparse structure matrices $\mathbf{C}_{11}, \mathbf{C}_{12}, \ldots, \mathbf{C}_{1b_1}$ where $s = \sum_{i=1}^{b_1} n_{si}$. (3) The tensor $\mathbf{C}_2 \in \mathbb{R}^{b_2 \times n_t \times n_t}$ stacked by $b_2$ dense structure matrices $\mathbf{C}_{21}, \mathbf{C}_{22}, \ldots, \mathbf{C}_{2b_2}$. (4) Marginal distributions of $b_1$ sparse matrices $\mathbf{p} = [\mathbf{p}_1 \mid \mathbf{p}_2 \mid \cdots \mid \mathbf{p}_{b_1}] \in \mathbb{R}^s$ where $\mathbf{p}_i \in \mathbb{R}^{n_{si}}$. (5) Marginal distributions of $b_2$ dense matrices $\mathbf{q} \in \mathbb{R}^{b_2 \times n_t}$. (6) $\mathbf{g} \in [1 : b_1]^s$ where $\mathbf{g}[a]$ is the graph index of node $a$ in the giant graph $\mathbf{C}_1$. (7) Trade-off $\alpha \in (0, 1)$, entropy $\epsilon > 0$. Max iterations $T$, tolerance $tol$

**Operator :** @: @ in pytorch, matrix/tensor multiplication. $*$: elementwise multiplication. $.view(\ldots)$: pytorch tensor $.view(\ldots)$. $[:, \text{ind}, :]$: pytorch tensor slice.

**Output:** Optimal transport plans $\mathbf{T} \in \mathbb{R}^{b_2 \times s \times n_t}$ of and $FGW$ distances between $b_1 b_2$ pairs

1  Initialize $\mathbf{T} \leftarrow \mathbf{p}.view(s, 1) * \mathbf{q}.view(b2, 1, n_t)$;
2  Elementwise function: $f(\mathbf{C}_1) \leftarrow \mathbf{C}_1^{\odot 2}, h(\mathbf{C}_1) \leftarrow \mathbf{C}_1, f(\mathbf{C}_2) \leftarrow \mathbf{C}_2^{\odot 2}, h(\mathbf{C}_2) \leftarrow \mathbf{C}_2$;
3  Precompute constants:
   $R \leftarrow f(\mathbf{C}_1)@\mathbf{p}.view(s, 1)@\mathbf{1}_{n_t}^\top + \mathbf{1}_{n_s}.view(s, 1)@\mathbf{q}.view(b_2, 1, n_t)@f(\mathbf{C}_2)$;
4  Define gradient operator $\nabla_{\mathbf{T}} \leftarrow 2\alpha(R - 2h(\mathbf{C}_1)@\mathbf{T}@h(\mathbf{C}_2)) + (1 - \alpha)\mathbf{M}$;
5  **for** $t \leftarrow 1$ **to** $T$ **do**
6      $\mathbf{T}_{\text{prev}} \leftarrow \mathbf{T}$;
       /* Row projection                                         */
7      $\mathbf{T} \leftarrow \mathbf{T} * \exp(-\nabla_{\mathbf{T}}/\epsilon)$ // Update
8      $\mathbf{S}_r = \mathbf{p}/(\mathbf{T}.sum(-1))$ // Row scaling factor
9      $\mathbf{T} \leftarrow \mathbf{S}_r.view(b2, s, 1) * \mathbf{T}$ // Normalize rows
       /* Column projection                                   */
10     $\mathbf{T} \leftarrow \mathbf{T} * \exp(-\nabla_{\mathbf{T}}/\epsilon)$ // Update
       /* Sum the elements from the same sparse matrix along the 2nd dim      */
11     group_sum = scatter_sum($\mathbf{T}, \mathbf{g}, dim = 1$)
       /* Broadcast to each group and get columnm scaling factor           */
12     $\mathbf{S}_c = \mathbf{q}.view(b_2, 1, n_t)/\text{group\_sum}[:, \mathbf{g}, :]$
13     $\mathbf{T} \leftarrow \mathbf{S}_c * \mathbf{T}$ // Normalize columns
14     **if** $t \mod 10 = 0$ **then**
15         Compute error: $err \leftarrow \|\mathbf{T} - \mathbf{T}_{\text{prev}}\|_F$;
16         **if** $err < tol$ **then**
17             **break**;
18         **end**
19     **end**
20 **end**
21 **return** $\mathbf{T}$ and corresponding $FGW$ distances

## B.2 Backward

With $L(\mathbf{C}_{1i,k}, \mathbf{C}_{2j,l}) = (\mathbf{C}_{1i,k} - \mathbf{C}_{2j,l})^2$, the gradients of 1-to-1 FGW w.r.t. the input elements are

$$\frac{\partial FGW}{\partial \mathbf{M}} = (1 - \alpha)\mathbf{T}$$

$$\frac{\partial FGW}{\partial \mathbf{C}_1} = 2\alpha\mathbf{C}_1 \odot (\mathbf{p}\mathbf{p}^\top) - 2\alpha\mathbf{T}\mathbf{C}_2\mathbf{T}^\top$$

$$\frac{\partial FGW}{\partial \mathbf{C}_2} = 2\alpha\mathbf{C}_2 \odot (\mathbf{q}\mathbf{q}^\top) - 2\alpha\mathbf{T}^\top\mathbf{C}_1\mathbf{T}$$

$$\frac{\partial FGW}{\partial \alpha} = \left\langle \left[\mathbf{C}_1^{\odot 2}\mathbf{T}\mathbf{1}_{N_2}\mathbf{1}_{N_2}^\top + \mathbf{1}_{N_1}\mathbf{1}_{N_1}^\top\mathbf{T}\left(\mathbf{C}_2^{\odot 2}\right)^\top - 2\mathbf{C}_1\mathbf{T}\mathbf{C}_2^\top\right], \mathbf{T} \right\rangle - \langle \mathbf{T}, \mathbf{M} \rangle_F$$

$$\frac{\partial FGW}{\partial \mathbf{p}} = \alpha\mathbf{C}_1^{\odot 2}\mathbf{p} \qquad \frac{\partial FGW}{\partial \mathbf{q}} = \alpha(\mathbf{C}_2^{\odot 2})^\top\mathbf{q}.$$

The backward computations of $b_1 b_2$ FGW distances can be parallelized via similar techniques introduced in parallel FGW computation described in the forward process of our BAPG layer (Section B.1).

## C   More Experiments

### C.1   Implementation Details

The implementation is based on Pytorch2.5 [52] and PyG2.6 [17]. All experiments are conducted on Ubuntu 22.04 server equiped with an RTX 4090 GPU and Intel Xeon Gold 6240C CPU. For FGW [59, 39], we implement an extremely efficient version from scratch (Section B) instead of using Python Optimal Transport toolbox [20].

Many studies report test performance based on the epoch that achieves the best performance on the validation set. However, in the context of NCD, selecting the epoch based on clustering accuracy over unlabeled validation samples would require knowledge of the ground truth new-class labels, which violates the NCD assumption. Alternatively, using old-class accuracy on the validation set may reinforce model bias toward known classes, potentially harming performance on new classes. Therefore, we report results from the final training epoch. In addition to the

Table 7: The split information of four GLNCD datasets

| Dataset | # train | # test | # all |
| --- | --- | --- | --- |
| ENZYMES | 420 | 120 | 600 |
| MalNet-Tiny | 3500 | 1000 | 5000 |
| REDDIT12K | 8350 | 2386 | 11929 |
| CIFAR10 | 35000 | 10000 | 60000 |

three baseline methods designed in Section 3.2, we also implement a K-means baseline: after pretraining the GNN encoder with GraphCL, we apply K-means directly to the GNN representations of the unlabeled training samples and old-class test samples, and report the corresponding clustering accuracies. For all experiments, we use the AdamW optimizer [45] and cosine annealing scheduler [44] with warmup.

Following common practice in visual NCD [80, 78, 27], we first determine the hyperparameters for AutoNovel on a given dataset. These hyperparameters are then inherited by NCL and DualRS on the same dataset, and only the hyperparameters that differ from AutoNovel are subsequently tuned for these two methods. ProtoFGW-NCD has a fundamentally different architecture from the above baselines, and its hyperparameters are therefore not influenced by AutoNovel. The hyperparameter values or search spaces of these mehods are presented in Table 8.

### C.2   Benchmarking Our BAPG Layer Implementation

Section B introduces our BAPG layer for efficient, differentiable FGW distance computation, which leverages `torch_sparse` and `torch_scatter` to parallels pairwise FGW computations between $b_1$ sparse matrices and $b_2$ dense matrices. Compared to the BAPG implementation in POT [6], our improved version supports sparse matrices, parallelized iterative solving, and efficient automatic differentiation. These enhancements significantly promote the broader application of the FGW distance in graph-level machine learning. In this section, we compare our parallel BAPG implementation with POT on a series of synthetic attributed graph datasets.

Graph sizes considered are 20, 50, 100, 200, 500, and 1000 nodes. For each size, we generate 1000 Contextual Stochastic Block Model (CSBM) graphs [14, 48]. Prototype graphs are typically used to represent large-scale topological relationships; thus, each node in a prototype graph corresponds to a block or community of nodes. While no universally accepted formula exists for determining the optimal number of communities in a network, empirical studies suggest that the number of communities typically grows sub-linearly with the number of nodes $N$, often approximated as $O(N/\log N)$, especially in scale-free or real-world networks where community sizes follow a power-law distribution [51, 3]. Therefore, for graphs of size $N$, we use 10 prototype graphs of size $N/2 \log N$ to capture patterns in the dataset, and the dataset is denoted CSBM-$N$-10. In Figure 6, we display 10 prototype graphs used to generate 1000 graphs in CSBM-100-10 dataset. Commonly used batch sizes in graph learning are 64, 128, 256, and 512. We evaluate the efficiency of different

---

[6]https://pythonot.github.io/gen_modules/ ot.gromov.html#ot.gromov.BAPG_fused_gromov_wasserstein

Table 8: Hyperparameter values and search spaces of GLNCD methods

| Group | Hyperparameter | Value or Search Space |
|---|---|---|
| **Common Hyperparameters** | | |
| Optimization | Learning rate | [0.001, 0.005, 0.01, 0.05, 0.1] |
| | Dropout | [0.1, 0.2, 0.3, 0.4, 0.5, 0.6, 0.7, 0.8] |
| | Weight decay | [0.0, 5e-5, 1e-4, 5e-4, 1e-3, 5e-3, 1e-2] |
| | Cosine warmup steps | [2, 5, 10] |
| | Batch size | [64, 128, 256, 512] |
| | Max epochs | [20, 50, 100, 300] |
| Neural Network Arch. | GNN encoder layer | [2, 3, 4, 5, 6] |
| | Hidden dimension | [32, 64, 128, 256] |
| | has_residual | [False, True] |
| | has_ffn | [False, True] |
| | Normalization | [batchborm, None] |
| Graph SSL | Node droprate | [0, 0.1, 0.2, 0.3, 0.4, 0.5, 0.6] |
| | Temp. for contrastive loss | [0.1, 0.3, 0.5, 0.7, 0.9, 1.1] |
| **GLNCD method Hyperparameters** | | |
| All baselines | Encoder pooling readout | ['mean', 'add', 'max'] |
| AutoNovel | Topk in RS | [5, 10, 15] |
| | Rampup length | [10, 50, 80, 150, 300] |
| | Rampup coefficient | [1.0, 5.0, 25., 50.] |
| NCL | Labeled NCL loss weight | [0.2, 1] |
| | Unlabeled NCL loss weight | [0.2, 1] |
| | Queue length | [200, 2000] |
| DualRS | Memory bank length | [256, 512, 1024] |
| ProtoFGW-NCD | Epsilon | [0.01, 0.05, 0.07, 0.11, 0.15, 0.19] |
| | Prototype node feature std. | [0.5, 1.] |
| | # prototype graphs | range(10, 130, 10) |
| | # prototype graph nodes | 20 |

BAPG implementations in comparing all 1000 graphs with the 10 prototype graphs under various batch sizes.

The average time for computing the FGW distance between a batch of graphs and all 10 prototype graphs, under different batch sizes across various datasets, is shown in Table 6. The total time required to traverse each entire dataset is presented in Figure 4. As summarized in Table 6, our parallel BAPG solver delivers dramatic runtime improvements over the POT implementation across all batch sizes and problem scales. While POT's runtime grows roughly linearly with batch size (e.g., from $\approx$8.9 s at $B$=64 to $\approx$67.3 s at $B$=512 on CSBM-20-10), our approach maintains a nearly constant per-batch cost ($\approx$0.02–0.04 s), yielding speedups that increase from $\approx$250$\times$ to $\approx$2070$\times$ as $B$ grows. Furthermore, the degree of acceleration decreases as the graph size $N$ increases—exceeding 400$\times$–2000$\times$ for small-to-medium sizes ($N$$\leq$100) at $B$=512, yet still achieving 3$\times$–14$\times$ for large-size graphs ($N$=500–1000). These findings demonstrate that our technique effectively amortizes overhead and exploits parallelism for batches of FGW problems, offering exceptional throughput for the tasks of small- and medium-size graphs while retaining nontrivial gains even in large-graph settings. This is also supported by the epoch time comparison displayed in Figure 4.

# D Limitations

As the first work to consider graph-level NCD, *this paper aims to introduce the new task of GLNCD and examine whether existing visual NCD methods can be effectively adapted by simply replacing their components with graph-domain counterparts*. To address this question, we adapt three classic NCD methods from computer vision to establish GLNCD baseline approaches (Section 3.2) and evaluate their performance on four newly designed GLNCD datasets spanning different domains (Section 3.1).

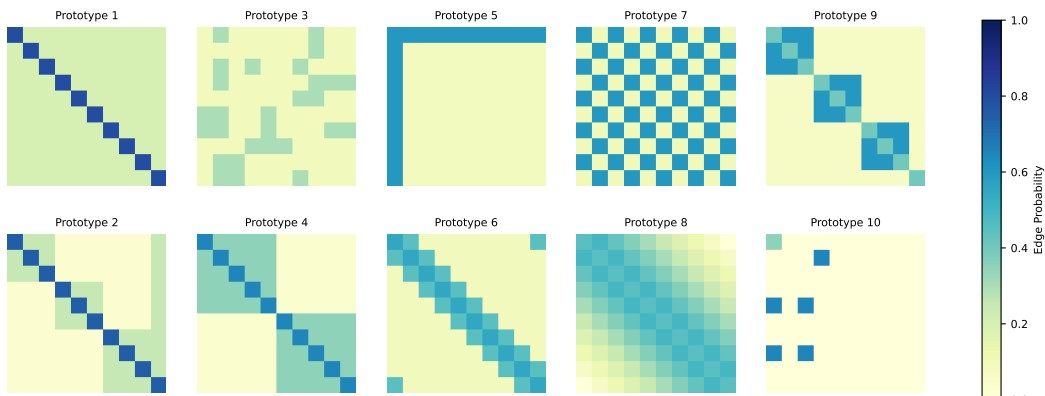

Figure 6: The prototype graphs used to synthesize the 1000 graphs in CSBM-100-10 dataset.

Our experimental results and analysis (Section 4) clearly indicate a negative answer, suggesting that current graph SSL methods and ranking statistics are insufficient in capturing structural information within graphs, thereby leading to low GLNCD performance. Although we do not adapt the most recent visual NCD methods, the limitations we observe from direct transfer can be generalized to them. This is because these recent visual NCD methods would still rely on existing graph SSL techniques for representation learning and employ pseudo-labeling strategies that neglect graph structure to train the new-class head. Additionally, despite the development of our parallel BAPG solver (Section B), which significantly improves computational efficiency in solving FGW for graph learning—achieving an impressive 2070× speedup on the CSBM-20-10 dataset where graphs have 20 nodes—the experimental results (Section C.2) show that the acceleration drops to only about 3.3× on larger graphs (CSBM-1000-10). Future work may focus on analyzing the causes of reduced speedup and improving the implementation, or exploring alternatives to FGW that more efficiently exploit graph structure to enhance both graph SSL and pseudo-labeling strategies.

