# OpenReview forum: "GLNCD: Graph-Level Novel Category Discovery"
_NeurIPS.cc/2025/Conference — NeurIPS 2025 poster_

### Official Review · Reviewer_adDu · 2025-06-15

**Clarity:** 2
**Significance:** 3
**Originality:** 3
**Rating:** 3
**Confidence:** 4

**Summary:**

The authors explored the task of discovering new classes in the graph-level tasks. They begin by analyzing the performance of existing CV methods on graph data, highlighting their inability to effectively utilize structural information. To address this, the authors introduced the Gromov-Wasserstein distance and designed a method that can simultaneously encode both structural and features information by calculating the distance to many learnable prototype graphs. The authors claimed that experiments showed the effectiveness of the proposed method.

**Questions:**

Please see the Weaknesses.

**Ethical Concerns:**

["NO or VERY MINOR ethics concerns only"]

**Final Justification:**

As I mentioned in my earlier comments, I do not deny the innovation and contribution of this paper, nor do I insist that it should be rejected. However, the proposed method achieves only marginal improvements over approaches that do not incorporate graph structure. This raises two important questions: 1) whether the authors have effectively leveraged the structural information, and 2) whether such structural information should, in fact, be considered for this task. Considering these issues, along with the lack of responses to new comments of all reviewers, I have maintained my score of 3.

**Limitations:**

Yes.

**Paper Formatting Concerns:**

NA.

**Quality:**

2

**Strengths And Weaknesses:**

Strengths:

- The authors explore a novel and valuable problem, which is discovering new classes in graph-level tasks.

- The proposed method is innovative. The authors design multiple prototype graphs and calculate the distribution distance between graphs and prototype graphs to obtain node representations. This idea provides valuable insights to the readers.

- The authors' analysis of CV methods is valuable.

Weaknesses:

- The primary drawback of this paper, in my view, is that the proposed method does not seem to work. In Tables 3 and 4, the improvements of the proposed method compared to existing CV methods are too marginal. Particularly in Table 4, the results for new classes show that the proposed method achieves the same rank of 2 as AutoNovel, indicating no improvement.

- The organization of Section 5 could be improved to enhance clarity. The authors might consider providing an overview of the entire method first, then detailing each part individually. Additionally, I suggest including more explanations to clarify the rationale behind each step, which might make it easier for readers to follow. While I understand the challenge of explaining thoroughly within limited space, the authors could make more references to the appendix and provide additional details there.

- The motivation for selecting the Gromov-Wasserstein distance is missing. There are many existing methods to measure structural similarity. The authors should explain why Gromov-Wasserstein was chosen over others.

- The experiments lack a report on the results of the balance weight $\alpha$ between structure and feature. It is a learnable parameter, and I believe the authors should provide the learned values to illustrate the extent to which the proposed method leverages structural information.

- I think the authors need to report the results of the proposed method under the settings of Figures 1 and 2 for a comprehensive evaluation of the proposed method.

---

> ### Author Rebuttal · Authors · 2025-07-31
>
> We sincerely appreciate the reviewer's thoughtful feedback and constructive suggestions, which have helped us identify areas for improvement in both the presentation and technical rigor of our work. Below, we provide a point-by-point response to the reviewer's concerns.
>
> ## **Weakness 1**
>
> > The primary drawback of this paper, in my view, is that the proposed method does not seem to work. In Tables 3 and 4, the improvements of the proposed method compared to existing CV methods are too marginal. Particularly in Table 4, the results for new classes show that the proposed method achieves the same rank of 2 as AutoNovel, indicating no improvement.
>
> As a pioneering work in this field, the primary goal of this paper is to establish comprehensive benchmarks and methodologies, provide in-depth analysis of their performance, and offer valuable insights to the community.
>
> Our ProtoFGW-NCD is primarily designed to validate the argument at the end of Section 4 (i.e.,the lack of structural information causes methods like G-AutoNovel to fail), rather than to propose an optimal method. To strictly control other variables for fair comparison, ProtoFGW-NCD only inserts a BAPG layer into the representation output layer of G-AutoNovel to incorporate structural information, while keeping all other key components (e.g., training loss functions, augmentation strategies) identical.
>
> Regarding the experimental results, in Tables 3 and 4, ProtoFGW-NCD achieves the top rank in performance for old classes and new classes, except for the avg. rank of New ACC in Table 4, where it ties with AutoNovel. However, this does not imply no improvement. On the three inherently graph-structured datasets (ENZYMES, MalNet-Tiny, and REDDIT12K), ProtoFGW-NCD improves over AutoNovel by **7.99%**, **1.47%**, and **4.33%**, respectively. Given that ProtoFGW-NCD differs from G-AutoNovel only by a single BAPG layer, we argue that such improvements already indicate meaningful progress.
>
>  As emphasized in the Limitations section, future work should explore more efficient modules for mining structural information to achieve better performance, but that is not the primary goal or core contribution of this paper.
>
> ## **Weakness 2**
> >
> >The organization of Section 5 could be improved to enhance clarity. The authors might consider providing an overview of the entire method first, then detailing each part individually. Additionally, I suggest including more explanations to clarify the rationale behind each step, which might make it easier for readers to follow. While I understand the challenge of explaining thoroughly within limited space, the authors could make more references to the appendix and provide additional details there.
>
> Thank you for your constructive suggestion. We will reorganize the content according to your advice in future revisions. Here, we provide some additional explanations for Section 5 to enhance clarity.
>
> **Motivation of Section 5.** At the end of Section 4, we propose the viewpoint that the lack of structural information affects both the SSL part (i.e., GraphCL) and the RS part of AutoNovel, leading to its failure. To validate this claim, we designed, ProtoFGW-NCD, a method that is identical to AutoNovel except for one structural information component. It injects structural information into SSL and RS via a BAPG layer, without requiring additional loss terms or more sophisticated strategies.
>
> **Section 5.1: ProtoFGW-CL, Minimal GraphCL Variant with Explicit Structural Information.**
> Existing Graph SSL methods more complex than GraphCL often focus on augmentation strategies and self-supervised loss design. While some incorporate structure, they still do not consistently outperform GraphCL [20, 58]. Thus, we chose a different approach: directly adding a BAPG layer after the GraphCL backbone to inject structural information into the representations. This design allows our method to serve as an extremely simple variant of GraphCL for the comparison study with AutoNovel.
>
> **Prototypes in the BAPG Layer.** SSL requires comparing a set of $b_1$ graphs $\mathcal{G}_1$ with another set of$b_2$ graphs $\mathcal{G}_2$. Graph comparison tools that explicitly incorporate structural information include GED and FGW, among which we selected FGW (see response to **Weakness 3** for the behind rationale). Existing differentiable FGW layer requires brute-force looping over$b_1 \times b_2$ pairs, making it difficult to leverage modern GPU parallelism: primarily because comparing graphs of varying sizes is inherently non-parallelizable. We found that if the graphs in $\mathcal{G}_2$ share the same size, FGW can be efficiently parallelized (see Appendix B and C.2 of our paper). In practice, this is hard to guarantee, so we introduce a set of fixed-size prototype graphs as intermediaries, and then compare variable-sized graphs by comparing their relationships with these prototypes.
>
> **Sec. 5.2: FGW-RS, Minimal RS Variant with Explicit Structural Information.**
> Since the representations output by the BAPG layer already contain structural information, feeding them directly into traditional RS yields a simple RS comparison method. Had we used vector similarity thresholds instead of RS to generate pseudo-labels, ProtoFGW-NCD (FGW-RS) would no longer serve as a proper and fair comparison to AutoNovel (RS). Moreover, once the representations incorporate structural information, the advantages of RS over vector similarity [21] should manifest as expected.
>
> **Sec. 5.3: ProtoFGW-NCD comprised of ProtoFGW-CL and FGW-RS.**
> Combining the two is natural, as both only require a same BAPG layer. This section primarily outlines the entire pipeline and highlights the advantages of the BAPG layer over the previous differentiable FGW layer, TFGW layer [55].
>
> ## **Weakness 3**
> >
> > The motivation for selecting the Gromov-Wasserstein distance is missing. There are many existing methods to measure structural similarity. The authors should explain why Gromov-Wasserstein was chosen over others.
>
> In addition to (F)GW distance, there are other structural similarity measures such as GED, graph kernels, and spectral distance.
>
> - **Compared to these methods, FGW's key advantage lies in its differentiability via implicit differentiation, which enables seamless integration into modern end-to-end systems**. Other methods, including but not limited to GED, spectral distance, Weisfeiler-Lehman kernel, and random walk kernel, are non-differentiable, and it is challenging to render them differentiable.
>
> - **FGW couplings provide node-level interpretability for graph comparison**.
> - **We find that FGW supports batch-to-batch parallelization on GPUs (see Appendix B for our parallel design)**. While other methods can be (approximately) solved with similar complexity to FGW for a single 1-to-1 graph comparison, they cannot achieve the same level of batch parallelism.
>
> ## **Weakness 4**
> >
> > The experiments lack a report on the results of the balance weight $\alpha$ between structure and feature. It is a learnable parameter, and I believe the authors should provide the learned values to illustrate the extent to which the proposed method leverages structural information.
>
> We conducted experiments with different numbers of prototype graph on ENZYMES, to observe the learned balance weight values (i.e., the leverage of structural information) under different scenarios.
>
> From **Tables R1** and **R2**, we found that:
>
> - In all cases, our method learned non-negligible $\alpha$, indicating that the proposed approach indeed leverages structural information.
> - Different backbones and prototype numbers influence $\alpha$, with the backbone having a significant impact.
>
> ***Table R1: The performance on ENZYMES with GCN+ backbone. Results are averaged over 10 runs.***
>
> | **K** | 30 | 50 | 70 | 90 | 120 | 150 |
> |---|---|---|---|---|---|---|
> | **Old ACC** | 68.00 | 69.50 | 65.33 | 66.17 | 70.33 | 68.33 |
> | **New ACC** |44.43 | 45.62 | 48.05 | 48.00 | 47.76 | 46.81 |
> | **Learned $\alpha$** |0.34 | 0.45 | 0.53 | 0.56 | 0.54 | 0.60 |
>
> ***Table R2: The performance on ENZYMES with GIN+ backbone. Results are averaged over 10 runs.***
>
> | **K** |30 | 50 | 70 | 90 | 120 | 150 ||||
> |---|---|---|---|---|---|---|---|---|---|
> | **Old ACC** | 73.33 | 64.67 | 71.83 | 72.83 | 68.33 | 71.17 ||||
> | **New ACC** | 37.62 | 38.14 | 38.19 | 35.24 | 38.86 | 37.95||||
> | **Learned $\alpha$** | 0.22 | 0.26 | 0.25 | 0.23 | 0.25 | 0.26 ||||
>
> ## **Weakness 5**
> >
> > I think the authors need to report the results of the proposed method under the settings of Figures 1 and 2 for a comprehensive evaluation of the proposed method.
>
> Given the restrictions on the rebuttal format imposed by the official Program Chairs, we are unable to present figures such as Figure 1 and 2. Instead, we selected ten consecutive and equally spaced time points and presented the changes in New ACC and MCC (i.e., pseudo-label quality) in **Table R3**.
>
> In the early stages of training, as MCC improved, New ACC also showed significant improvement. However, in the later stages, after MCC stabilized, New ACC continued to exhibit slight improvements.
>
> ***Table R3: The training dynamics on ENZYMES with GCN+ backbone. Results are averaged over 10 runs.***
>
> |Epoch|0|20|40|60|80|100|120|140|160|180|200|
> |---|---|---|---|---|---|---|---|---|---|---|---|
> |MCC|0.147|0.167|0.200|0.200|0.223|0.244|0.246|0.247|0.245|0.248|0.247|
> |New ACC|0.383|0.409|0.404|0.405|0.411|0.446|0.447|0.419|0.404|0.457|0.474|
>
> ## References
>
> - [20] Architecture matters: Uncovering implicit mechanisms in graph contrastive learning, NeurIPS 2023.
> - [21] AutoNovel: Automatically Discovering and Learning Novel Visual Categories, TPAMI, 2022.
> - [55] Template based Graph Neural Network with Optimal Transport Distances NeurIPS 2022.
> - [58] SimGRACE: A Simple Framework for Graph Contrastive Learning without Data Augmentation, WWW 2022.

---

> > ### Comment · Reviewer_adDu · 2025-08-04
> >
> > Thanks for the response. The issues raised in W2-5 have been well addressed. However, I do not fully agree with the response to W1. As the authors themselves mentioned, they are pioneers in this field. In my view, considering that other competing methods do not specifically adapt to structured data, their approach should outperform existing works. It is not sufficient to justify minor improvements merely because the work is pioneering, especially given the high standards of NeurIPS.
> >
> > Therefore, I maintain my current score.  That said, I acknowledge the authors' responses and the perspectives of the other reviewers and AC. I do not insist that the paper should be rejected.
> >
> > Furthermore, if the authors can demonstrate a more substantial difference in performance before and after adapting existing NCD methods to graph structures (i.e. the proposed BAPG layer), or show more advanced results from their proposed method, I would be open to reassessing my score.

---

### Official Review · Reviewer_n3Hv · 2025-07-02

**Clarity:** 3
**Significance:** 4
**Originality:** 4
**Rating:** 5
**Confidence:** 4

**Summary:**

This study investigates the traditional Novel Class Discovery problem in the context of graph-level classification. The authors first introduce several graph datasets suitable for NCD evaluation and design three Graph-Level NCD baselines based on existing image-based NCD methods. Subsequently, they analyze these baselines and identify key limitations that conventional visual NCD designs fail to effectively incorporate graph structural information, leading to suboptimal performance on graph data. To address this issue, the authors propose a novel approach based on optimal transport that integrates graph structural information.

**Questions:**

See in Weaknesses.

**Ethical Concerns:**

["NO or VERY MINOR ethics concerns only"]

**Final Justification:**

The author's response largely addressed my concerns.

**Limitations:**

Yes

**Quality:**

3

**Strengths And Weaknesses:**

Strengths
1. This work appears to be the first to explore NCD in the context of graph-level classification. Potential applications, such as novel drug molecule discovery, could benefit from this research.
2. The experimental evaluation and analysis of the baselines reveal unique challenges specific to NCD on graph-structured data.
3. The authors design and implement an efficient and differentiable Fused Gromov-Wasserstein distance module, which facilitates the application of FGW in graph-level tasks.

Weaknesses
1. The study employs a relatively limited number of datasets and baselines.
2. The discussion on the challenges of Representation Similarity in graph data (Section 4.2) is somewhat unclear. A more detailed, point-by-point analysis with additional elaboration would strengthen this section.
3. The meaning of the "Avg. Rank/All" column in Tables 3 and 4 is not explicitly explained. Does it represent the average ranking across all preceding columns?
4. The proposed BAPG layer bears similarity to the method presented in [1], but a detailed comparison between the two is lacking.

[1] Template Based Graph Neural Network with Optimal Transport Distances, NeurIPS 2022.

---

> ### Author Rebuttal · Authors · 2025-07-31
>
> We sincerely thank the reviewer for the thoughtful and constructive feedback on our manuscript. The reviewer's insightful comments have helped us identify areas for clarification and improvement. We appreciate the recognition of our work's novelty in exploring NCD for graph-level classification and the potential applications in domains like drug discovery. Below, we provide a point-by-point response to address the reviewer's concerns
>
> ## **Weakness 1**
>
> > The study employs a relatively limited number of datasets and baselines.
>
> **Limited Datasets**:
> As discussed in Sec. 3.1, while there are over 120 public graph-level datasets, few are naturally suited for multi-class graph classification beyond CV-derived ones (e.g., CIFAR10, MNIST [10], Letter-high, COIL-RAG [42]). Since CV-to-graph conversion is not popular and underperforms CNNs, we retained only CIFAR10 (Graph). Truly suitable GLNCD benchmarks are scarce. Furthermore, we excluded large-scale datasets like ogbg-ppa [R1] since Graph SSL, a key component of GLNCD, currently lacks established scaling laws for large datasets [R2]. These fundamental challenges in large-scale GLNCD scenarios, along with scalability issues, warrant further investigation and are deferred to future work.
>
> **Limited Baselines**:
> As outlined in Appendix D, we conduct an initial exploration of GLNCD, addressing two key questions: **(1)** whether CV-based NCD methods trivially generalize to graphs, and **(2)** what unique challenges GLNCD presents compared to traditional NCD.
>  Our baseline G-AutoNovel and controlled experiments (e.g., feature quality manipulation in Sec. 4) show that standard NCD approaches can not trivially generalize to graph, suggesting structure information loss as a key challenge. This key challenge is further verified by comparing G-AutoNovel with the carefully designed ProtoFGW-NCD.
>  While our baselines are not exhaustive, they cover the core components of GLNCD (Graph SSL + Pseudo-Labeling) and are sufficient to answer the two key questions.
>
> ## **Weakness 2**
> >
> > The discussion on the challenges of Representation Similarity in graph data (Section 4.2) is somewhat unclear. A more detailed, point-by-point analysis with additional elaboration would strengthen this section.
> >
> We provide additional clarification for Section 4.2 for better understanding here. The section will be updated in the revised manuscript accordingly.
>
> In Section 4.1, we propose two metrics to evaluate pseudo-label quality and compare AutoNovel (CIFAR-10) with our G-AutoNovel baseline. Figure 1(b, c) reveals that RS in G-AutoNovel fails to generate effective pseudo-labels, unlike in the image domain.
>
> Section 4.2 investigates **Why RS Fails in Graphs?**
>
> **Hypothesis 1**: RS failure may stem from poor GraphCL (SSL) representations, resulting in low-quality pseudo-labels.
> **Test**: Using SupCon loss as an oracle, we train encoders with three representation quality levels and evaluate pseudo-label quality and NCD performance.
>
> **Finding 1**: Figure 2(b, c) shows pseudo-label quality remains consistent across representation levels, rejecting **Hypothesis 1**.
> **Finding 2**: Despite similar pseudo-label quality, NCD performance varies significantly (Fig. 1a). We conclude:
>
> - Representation quality primarily affects pseudo-label *utilization* (e.g., learning clearer decision boundaries) rather than their inherent quality.
> - This phenomenon does not mean that representation quality does not affect pseudo-label quality. On CIFAR-10 (Image), as representation quality improves with training, pseudo-label quality also increases.
>
> **Key Insight**: RS (in G-AutoNovel) is inherently unsuitable for graphs as it cannot refine pseudo-labels even with improved representations.
>
> **Core point**: GraphCL and RS in G-AutoNovel underutilize graph structural information, harming *utilization* and *quality* respectively. And this leads to low NCD performance.
>
> In Section 5, we augment G-AutoNovel with a minimal structural-aware BAPG layer to get ProtoFGW-NCD. And we validate the above core point by comparing ProtoFGW-NCD with G-AutoNovel.
>
> ## **Weakness 3**
> >
> > The meaning of the "Avg. Rank/All" column in Tables 3 and 4 is not explicitly explained. Does it represent the average ranking across all preceding columns?
>
> Yes, it is the average ranking of the metrics in the Old ACC and New ACC columns (a total of 8 columns) across the entire dataset. The model ranking for each column is determined by sorting within that column.
>
> ## **Weakness 4**
> >
> > The proposed BAPG layer bears similarity to the method presented in [1], but a detailed comparison between the two is lacking.
>
> The TFGW layer proposed by [51] computes the FGW distances between the input graph (with node representations) and $K$ template graphs. These $K$ distances form a $K$-dimensional explicit structural representation of the input graph, which is then fed into subsequent classification or regression heads.
>
> **Motivation**: TFGW is designed to improve GNN performance in supervised classification and regression tasks. In contrast, our approach aims to address parallelized comparison between graphs with irregular sizes in self-/semi-supervised tasks.
>
> **Working Principle**: The forward pass of the TFGW layer employs the Conditional Gradient algorithm from [51], and does not support batch parallelism. In comparison, our BAPG layer adopts a variant of the Bregman Alternating Projected Gradient (BAPG) algorithm [55] parallelized by us. While the gradient formula of the FGW distance remains consistent for a single 1-to-1 FGW problem, TFGW does not support parallelized batch-to-batch backward passes, significantly limiting its applicability.
>
> **Computational Efficiency**: The TFGW layer does not natively support GPU computation and requires data transfer between host (CPU) and device (GPU). In contrast, the BAPG layer is implemented via PyTorch's `torch.autograd.Function` interface, eliminating CPU-GPU data transfers. Thanks to this and its parallelized design, our BAPG implementation achieves up to 2070x speedup over the implementation in the well-known POT library [15] (see Appendix C.2).
>
> **Impact on the Field**: Our BAPG layer alleviates the efficiency limitations of previous differentiable FGW operators (e.g., the TFGW layer), potentially enabling wider use of FGW distance-based techniques in modern end-to-end graph learning systems.
>
> ## References
>
> - [10] Benchmarking Graph Neural Networks, JMLR, 2023.
> - [42] TUdataset: A collection of benchmark datasets for learning with graphs, ICLRW 2022.
> - [51] Optimal Transport for structured data with application on graphs, ICML 2019.
> - [55] Template Based Graph Neural Network with Optimal Transport Distances, NeurIPS 2022.
> - [15] POT: Python Optimal Transport, JMLR, 2021.
> - [R1] Open Graph Benchmark: Datasets for Machine Learning on Graphs, 2020.
> - [R2] Do Neural Scaling Laws Exist on Graph Self-Supervised Learning?, 2024.

---

> > ### Comment · Reviewer_n3Hv · 2025-08-02
> >
> > Thank you for your reply. My issue has been resolved, and I have updated the score.

---

### Official Review · Reviewer_4CRM · 2025-07-03

**Clarity:** 3
**Significance:** 4
**Originality:** 3
**Rating:** 4
**Confidence:** 3

**Summary:**

This paper introduces Graph-Level Novel Category Discovery (GLNCD), extending graph classification to open-world scenarios where models must discover novel graph categories without supervision. The authors propose ProtoFGW-NCD, addressing limitations of adapted visual NCD methods through two components: ProtoFGW-CL and FGW-RS. A parallel BAPG layer enables efficient FGW computation. Experiments on four diverse datasets show consistent improvements over adapted visual NCD baselines.

**Questions:**

1. How sensitive is performance to the number of prototype graphs K?

2. How does the α parameter in the FGW distance affect performance across different graph types and domains?

**Ethical Concerns:**

["NO or VERY MINOR ethics concerns only"]

**Final Justification:**

The author's response has effectively addressed my concerns regarding the technical details, and I think GLNCD represents a potentially valuable new task. Therefore, I maintain my original score.

However, after reading Reviewer adDu's response, I acknowledge the valid concerns raised about performance. I also note that the authors did not provide any response to Reviewer adDu's comments, leaving these important performance-related questions unaddressed. In light of this, I have adjusted my confidence level to 3, reflecting my support for the work's novelty while also respecting the unresolved concerns raised by other reviewers.

**Limitations:**

See Weakness

**Paper Formatting Concerns:**

None identified

**Quality:**

4

**Strengths And Weaknesses:**

**Strengths**:
1. The transition from closed-world to open-world graph classification addresses a genuine need in applications. While node-level NCD has been explored in prior work [1], this appears to be the first systematic study of graph-level NCD, representing a meaningful contribution to the graph learning literature.


2. The controlled experiments in Figures 1-2 effectively demonstrate that traditional ranking statistics fail on graph data even with improved representations. The insight that structural information loss, rather than representation quality, is the bottleneck provides clear motivation for the proposed FGW-based approach.


3. The authors develop a differentiable BAPG layer that enables efficient batch computation of FGW distances. The learnable prototype mechanism elegantly solves the variable-sized graph comparison problem.

**Weaknesses**:
1. While FGW distance is reasonable for graph comparison, the paper lacks a systematic comparison with other graph distance measures (e.g., Graph Edit Distance) or alternative approaches to structure-aware SSL. The choice appears more empirically driven than theoretically necessary, and the complexity-benefit trade-off is not thoroughly analyzed.

2. Given that FGW distance already provides a graph similarity measure, the insistence on using ranking statistics seems like conservative adherence to traditional NCD pipelines rather than optimal design. Why not directly use FGW distances for similarity judgment? The paper doesn't convincingly justify why ranking is still beneficial when operating on structure-aware FGW vectors.

3. The simple node/edge dropping augmentation seems naive for graph data. Have the authors considered more sophisticated graph augmentations that preserve semantic structure? The choice of augmentation strategy could significantly impact the quality of self-supervised learning.

[1] Jin, Y., Xiong, Y., Fang, J., Wu, X., He, D., Jia, X., ... & Yu, P. S. (2024, July). Beyond the Known: Novel Class Discovery for Open-World Graph Learning. In International Conference on Database Systems for Advanced Applications (pp. 117-133). Singapore: Springer Nature Singapore.

[2] Hou, Y., Chen, X., Zhu, H., Liu, R., Shi, B., Liu, J., ... & Xu, K. (2024, October). NC^2D: Novel Class Discovery for Node Classification. In Proceedings of the 33rd ACM International Conference on Information and Knowledge Management (pp. 849-859).

---

> ### Author Rebuttal · Authors · 2025-07-31
>
> We are truly grateful to the reviewer for the exceptionally thorough and insightful evaluation of our work. We sincerely appreciate the time and care you have taken to engage with our work so constructively. Your astute observations have been instrumental in refining key aspects of our methodology and presentation, as we detail in the following responses.
>
> ## **Weakness 1**
>
> > While FGW distance is reasonable for graph comparison, the paper lacks a systematic comparison with other graph distance measures (e.g., Graph Edit Distance) or alternative approaches to structure-aware SSL. The choice appears more empirically driven than theoretically necessary, and the complexity-benefit trade-off is not thoroughly analyzed.
>
> **Why not other graph distance measures?**
>
> - Compared to GED, graph kernels, and spectral distances, FGW's key advantage is its differentiability via implicit differentiation, enabling integration into modern end-to-end systems. The other methods including but not limited to GED, spectral distance, Weisfeiler-Lehman kernel, random walk kernel and spectral distances are non-differentiable, and it is hard to render these methods differentiable.
> - FGW couplings can provide node-level interpretability for graph comparison.
> - We find that FGW permits batch-to-batch parallelization on GPUs. While 1-to-1 approximated GED can match FGW's complexity, it cannot achieve batch parallelism.
>
> **Why not structure-aware SSL?**
>
> - Prior work ([20, 58]) shows structure-aware SSL methods don't consistently outperform GraphCL (used in our manuscript) across datasets.
> - The method proposed in Section 5 specifically aims to validate the hypothesis presented at the end of Section 4: that *structural information deficiency causes G-AutoNovel's failure*. To strictly isolate the variable of "introducing structural information," our comparative method should minimally modify the G-AutoNovel baseline. ProtoFGW-NCD achieves this by simply inserting a BAPG layer before the representation output to inject structural information. Alternative structure-aware SSL approaches typically require complex augmentations and auxiliary losses, leading to multiple architectural modifications that would compromise our controlled comparison.
> - From a purely performance perspective, replacing GraphCL with more sophisticated SSL methods appears promising, and we consider this an important direction for future work.
>
> **Complexity-benefit trade-off**
>
> The computational complexity for calculating FGW distance between graphs with $n$ and $m$ nodes is $O(n^2m + nm^2)$. In our implementation, we set the prototype graph size $m$=20 and $n$ depends on the dataset.
>
> Through parallelization of FGW computation between batches of $b_1$ and $b_2$ graphs, we achieved up to 2000x acceleration since all $b_1b_2$ pairs are parallelized. However, this acceleration essentially trades space for time, the total computational load remains unchanged. Therefore, from a purely practical standpoint, ProtoFGW-NCD is not optimal. As noted in Appendix D, our primary objective is to explore the novel GLNCD task and identify directions for improving baselines, rather than proposing the ultimate model. For validating our hypothesis in Section 4 (that structural information deficiency causes G-AutoNovel's failure), ProtoFGW-NCD serves as an appropriate choice. We reserve the development of more advanced solutions for future work.
>
> ## **Weakness 2**
>
> > Given that FGW distance already provides a graph similarity measure, the insistence on using ranking statistics seems like conservative adherence to traditional NCD pipelines rather than optimal design. Why not directly use FGW distances for similarity judgment? The paper doesn't convincingly justify why ranking is still beneficial when operating on structure-aware FGW vectors.
>
> **Why maintain traditional NCD pipelines?**
>
> At the end of Section 4, we proposed that the traditional G-AutoNovel baseline fails on graph data due to the insufficient exploitation of graph structure:
>
> - Lack of structural information in representations limits pseudo-label utilization
> - Readout vectors + Ranking statistics (RS) lacks structural information, and limits pseudo-label quality
>
> Sec. 5 is to design a controlled contrastive method to validate this hypothesis. To strictly isolate the variable of "introducing structural information" and observe its impact, we aimed to modify the G-AutoNovel baseline as minimally as possible while incorporating structural information. This ensures performance differences primarily stem from structural information injection rather than major architectural (including SSL loss) changes.
>
> Therefore, ProtoFGW-NCD in Section 5 was intentionally designed as a minimally modified version of G-AutoNovel. Its sole modification is inserting a BAPG layer before representation output, specifically to inject structural information (via FGW) into the final representations.
>
> **Why use ranking statistics (RS) ?**
>
> - Our hypothesis explicitly states that G-AutoNovel's failure stems not only from structural information deficiency in representations, but also from its RS mechanism (operating on simple readout vectors) failing to utilize structural information. To validate the second point, we needed to observe whether the RS mechanism would work more effectively when the input representations (signals to RS) contain structural information. Significant performance improvement would confirm the hypothesis.
>
> - Besides the first point, we know that direct similarity computation between vectors (e.g., $\mathbb{v}_1$ and $\mathbb{v}_2$ in Section 5.2) has been extensively shown to be less robust than RS in generating pseudo labels from vector pairs, as discussed in [21] (Discussion, p.4). Given that the structural information injected into representation vectors, RS is still very likely to outperform vector similarity measures.
>
> ## **Weakness 3**
>
> > The simple node/edge dropping augmentation seems naive for graph data. Have the authors considered more sophisticated graph augmentations that preserve semantic structure? The choice of augmentation strategy could significantly impact the quality of self-supervised learning.
>
> We have indeed investigated the potential benefits of sophisticated graph augmentations in graph-level SSL [20, 58]. According to reported results (e.g., Table 3 in [20]), despite more sophisticated augmentation designs, these methods failed to achieve superior performance across all datasets, suggesting that advanced augmentations do not always lead to the desired consistent gains.
>
> Furthermore, it's important to note that ProtoFGW-NCD was specifically designed as a minimally modified variant of G-AutoNovel to validate the hypothesis presented at the end of Section 4. Maintaining consistency in the augmentation strategy was therefore essential for controlled comparison.
>
> In summary, keeping our augmentation approach consistent with G-AutoNovel's implementation:
>
> - Allows for fair comparison with G-AutoNovel and solid verification of the effect of structural information.
>
> - Is supported by existing literature showing limited benefits of more complex augmentations
>
> ## **Question 1**
>
> > How sensitive is performance to the number of prototype graphs K?
>
> We add additional experiments on ENZYMES to investigate the sensitivity to hyperparameter $K$, as shown in **Tables RT1** and **RT2**. The results demonstrate that: *within the proper range, the model's performance remains relatively stable across different K values*
>
> ***Table R1: The performance sensitivity w.r.t. K on ENZYMES with GCN+ backbone. Results are averaged over 10 runs.***
>
> | **K** | 30 | 50 | 70 | 90 | 120 | 150 |
> |---|---|---|---|---|---|---|
> | **Old ACC** | 68.00 | 69.50 | 65.33 | 66.17 | 70.33 | 68.33 |
> | **New ACC** |  44.43 | 45.62 | 48.05 | 48.00 | 47.76 | 46.81 |
> | **Learned $\alpha\in [0,1]$** |  0.34 | 0.45 | 0.53 | 0.56 | 0.54 | 0.60 |
>
> ***Table R2: The performance sensitivity w.r.t. K on ENZYMES with GIN+ backbone. Results are averaged over 10 runs.***
>
> | **K** |  30 | 50 | 70 | 90 | 120 | 150 |  |  |  |
> |---|---|---|---|---|---|---|---|---|---|
> | **Old ACC** | 73.33 | 64.67 | 71.83 | 72.83 | 68.33 | 71.17 |  |  |  |
> | **New ACC** | 37.62 | 38.14 | 38.19 | 35.24 | 38.86 | 37.95  |  |  |  |
> | **Learned $\alpha\in [0,1]$** | 0.22 | 0.26 | 0.25 | 0.23 | 0.25 | 0.26 |  |  |  |
>
> ## **Question 2**
>
> > How does the $\alpha$ parameter in the FGW distance affect performance across different graph types and domains?
>
> $\alpha$ is a learnable parameter that adapts to the dataset's structure-feature-label relationships. It's initialized as 0.5, and adjusted through the training.
>
> As shown in **Tables RT1** and **RT2**, across all valid K settings, the model consistently learns moderate and well-calibrated $\alpha$ values that effectively balance structural and node feature information.
>
> We observe that when using GCN+ as backbone, the learned $\alpha$ values are generally higher than with GIN+, possibly correlating with GCN+'s significantly better New ACC performance.
>
> ## References
>
> - [20] Architecture matters: Uncovering implicit mechanisms in graph contrastive learning, NeurIPS 2023.
> - [58]  SimGRACE: A Simple Framework for Graph Contrastive Learning without Data Augmentation, WWW 2022.

---

> > ### Comment · Reviewer_4CRM · 2025-08-03
> >
> > Thank you for the authors' response regarding the technical details. I think GLNCD is a potentially interesting new task, and I maintain my borderline accept recommendation.

---

### Decision · Program_Chairs · 2025-09-17

**Decision:**

Accept (poster)

**Comment:**

This paper introduces Graph-Level Novel Category Discovery (GLNCD) and proposes ProtoFGW-NCD, which leverages prototype-based fused Gromov-Wasserstein distance and structure-aware pseudo-labeling to address the limitations of existing NCD methods on graphs. Reviewers agreed that the problem is timely and important, highlighting the novelty of formulating GLNCD and the efficiency of the proposed differentiable FGW module, but raised concerns about limited datasets, modest performance gains, and insufficient comparisons with alternative distance metrics. The authors clarified the motivation for FGW and provided additional explanations of their design choices, which addressed some concerns but left questions on scalability and evaluation breadth partly unresolved.